# Inner Information Analysis Algorithm for Deep Neural Network based on Community

**Guipeng Lan, Shuai Xiao,**\* **Meng Xi, Jiabao Wen & Jiachen Yang**
School of Electrical and Information Engineering
Tianjin University
Weijin Road Campus: No. 92 Weijin Road, Nankai District, Tianjin, China
`{lgp, xs611, ximeng, Wen_Jiabao, yangjiachen}@tju.edu.cn`

## Abstract

Deep learning has achieved advancements across a variety of forefront fields. However, its inherent 'black box' characteristic poses challenges to the comprehension and trustworthiness of the decision-making processes within neural networks. To mitigate these challenges, we introduce InnerSightNet, an inner information analysis algorithm designed to illuminate the inner workings of deep neural networks through the perspectives of community. This approach is aimed at deciphering the intricate patterns of neurons within deep neural networks, thereby shedding light on the networks' information processing and decision-making pathways. InnerSightNet operates in three primary phases, 'neuronization-aggregation-evaluation'. Initially, it transforms learnable units into a structured network of neurons. Subsequently, these neurons are aggregated into distinct communities according to representation attributes. The final phase involves the evaluation of these communities' roles and functionalities, to unpick the information flow and decision-making. By transcending focus on single-layer or individual neuron, InnerSightNet broadens the horizon for deep neural network interpretation. InnerSightNet offers a unique vantage point, enabling insights into the collective behavior of communities within the overarching architecture, thereby enhancing transparency and trust in deep learning systems.

## 1 Introduction

Deep learning has been instrumental in driving advancements across a range of domains, such as image recognition He et al. (2016); Lan et al. (2024), natural language processing Chowdhary & Chowdhary (2020), and reinforcement learning Moerland et al. (2023). Despite their impressive performance on diverse tasks, these networks typically operate as black-boxes with limited transparency. To address this issue, researchers have been actively investigating various strategies to demystify the inner workings of deep neural networks. These include visualization, model simplification, attribute specific features, etc. The focus of this work is to delve into the micro-structure of deep neural networks, particularly investigating the community partitioning among neurons. By identifying collaborative neural communities and analysing their role throughout the entire network, we can gain a deeper understanding of how neural networks process and make decisions internally.

The information flow and decision-making are controlled by complex non-linear interactions among parameters. Effective inner information analysis is crucial for identifying problems, correcting errors, and enhancing understanding. Inner information analysis can be defined as a method to describe the computational processes of deep neural networks in a way that is understandable to humans. In previous works, researchers have presented theories and tools from various perspectives, including weights, neurons, subnetworks, and latent representations. Regarding weights, ones train weight masks to determine which are important for specific tasks Wortsman et al. (2020); Csordás et al. (2021). This method is commonly used for network pruning Blalock et al. (2020); Frankle & Carbin (2018) to eliminate redundant neurons. Concerning neurons, a common evaluation method involves dataset-based analysis to identify neurons with maximum activation characteristics Zhou

---

\*Corresponding author

et al. (2014); Bau et al. (2017; 2018b). Additionally, establishing causal relationships between network behaviour and individual neurons by perturbing them is a common method Hod et al. (2021); Bau et al. (2018a). In terms of subnetworks, modularity is a universal principle that enables models to be understood by their independent parts Watanabe et al. (2018); Ruggeri et al. (2023). By analysing subnetworks, one aims to uncover hierarchical structures. Regarding latent representations, researchers infer the reasoning process of deep neural networks by analysing similarities with a group of learned 'prototypes' Alvarez Melis & Jaakkola (2018); Chen et al. (2019). Bengio et al. (2013); Koh et al. (2020) elucidate the behaviour of inner information by decoupling latent representations. Fong & Vedaldi (2018); Kim et al. (2018) explain the encoding of inner information by inducing images in conceptual datasets. Although the aforementioned works provide new insights into understanding deep neural networks, few studies have explored the behaviour and information flow from a holistic community perspective. This limitation restricts our ability to fully comprehend the inner mechanisms governing deep neural networks, particularly how these mechanisms contribute to the networks' decision-making processes. Consequently, further research is necessary to investigate the behaviour and information flow from a community-wide perspective.

To enhance understanding of neural networks, we propose InnerSightNet, a novel approach that examines networks from a community perspective rather than focusing on individual neurons or layers. InnerSightNet operates in three phases: neuronization (structuring learnable units into neurons), aggregation (grouping neurons into communities), and evaluation (analyzing their roles and interactions). This framework reveals intricate patterns of information flow and decision-making, offering deeper insights into the functions and interactions of these communities.

**Our contribution** We summarize the contribution of this work. **Algorithm** We propose a community-based inner information analysis algorithm for the transparency of deep neural networks, which broadens the perspective of deep neural network interpretation. We use the information representation of neurons to explore the community clustering effect in deep neural networks. In addition, this paper provides the mechanisms for analysing the role and function of the communities. **Applications** We apply InnerSightNet to commonly used structures in deep neural networks: linear neural networks and convolutional neural networks, and prove its effectiveness.

## 2 RELATED LITERATURE

The quest to demystify deep learning models has led to extensive research into various interpretability methods. We review in the field of deep learning interpretability, internal information analysis, community detection and the unique aspects that set our InnerSightNet apart from existing methods.

**Interpretability of Deep Learning:** Deep learning has achieved remarkable success across multiple domains; however, their 'black box' nature has prompted researchers to develop techniques aimed at improving their transparency. Visualization methods, such as Grad-CAM Selvaraju et al. (2017) and saliency maps Adebayo et al. (2018), have been widely used to highlight the regions that contribute most to the network's output, thus providing insights into the model's reasoning process. In additional, model simplification techniques, including pruning and quantization, have been proposed to reduce the complexity of neural networks, making them more interpretable. Feature attribution methods like LIME Zhao et al. (2021); Marvin et al. (2023) and SHAP Lundberg & Lee (2017); Bordt & von Luxburg (2023) further contribute to this effort by assigning importance scores to individual features, thereby elucidating how specific inputs influence the model's predictions.

**Internal Information Flow and Representation Analysis:** In addition to visualization and simplification, some work focuses on analyzing the internal information flow of neural networks. This type of analysis typically involves evaluating the importance of weights and neurons to identify the components that are crucial to performance. Weight masking techniques play a crucial role in network pruning and optimization by training specific weight masks to determine which weights are most important in a particular task Blalock et al. (2020); Frankle & Carbin (2018). In addition, neuron activation analysis helps to understand which parts have greatest impact on a specific task by detecting which neurons in the network exhibit the greatest activation. Latent representation analysis provides an understanding of how neural networks process and store information by studying the latent variables and eigenvectors within the network Bengio et al. (2013); Koh et al. (2020). Prototype inference helps reveal the classification and decision-making basis of the network by comparing its output with the prototype of known concepts Alvarez Melis & Jaakkola (2018); Chen et al. (2019).

**Community detection:** is a concept in network science Bedi & Sharma (2016), Elali & Rachid (2023), Goodley et al. (2024), aimed at identifying connected subgroups in a network, where the connections within these subgroups are denser than those with external nodes. Recently, community detection has been introduced into the analysis of GNN, becoming an innovative means of understanding the internal structure Sun et al. (2021); Su et al. (2022). These methods provide a new perspective for understanding the modularity of graph by identifying closely related groups of nodes that work together to achieve specific functions. Traditional community detection, such as modular optimization Que et al. (2015); Traag et al. (2019) or clustering methods Li et al. (2021); Van Lierde et al. (2019), has been improved and adjusted to adapt to high-dimensional, nonlinear features. The information exchange between neurons within a neural network can be conceptualized as a specialized unidirectional graph, where neurons serve as nodes and the information flow serve as directed edges. This inherent structure makes the application of community detection in the internal analysis of neural networks both logical and effective.

In summary, the existing methods advance our understanding of neural networks, they fail to capture the dynamic interactions and functional roles within neuron communities. InnerSightNet addresses these limitations, and enable a more nuanced exploration of community and information flow.

## 3 INNERSIGHTNET

InnerSightNet adaptively searches for community associations and explores information flow and decision-making. To achieve this, InnerSightNet neuronizes learnable units and aggregates neurons with similar roles as communities based on the input-output representations. InnerSight-Net represents an iterative algorithm that finds the best community allocation through continuous 'aggregation-evaluation'. InnerSightNet is divided into three steps: '**neuronization-aggregation-evaluation.**' The details are as follows. (We provide Theoretical background in Appendix: A.2)

### 3.1 INNERSIGHTNET: NEURONIZATION

We emphasize that structure plays a pivotal role in information flow and decision-making. We classify the fundamental structures into two categories: learnable units (convolutional layer, linear layers, etc.) and invariant non-linear units (normalization layers, pooling layers, activation functions, etc.). The invariant non-linear units are treated as a consistent non-linear function, unaffected by any alterations in the weights. Consequently, our study focus on the learnable units.

A linear layer consists of multiple hidden nodes, functioning as idealized neurons. In contrast, a convolutional layer, composed of kernels, outputs 2D representations, with each kernel analogous to a neuron. Based on the continuity of convolutional kernel outputs revealed by Bau et al. (2017), we quantify the correlation between the $p$-th kernel in layer $d$, $\kappa_p^d$, and the $q$-th kernel in layer $d+1$, $\kappa_q^{d+1}$, as shown in Eq. 1.

$$\delta(p,q) = \frac{1}{N} \sum_{i=1}^{N} \frac{KL_i(\kappa_p^{d^+}, \kappa_q^{d+1^+}) + KL_i(\kappa_p^{d^-}, \kappa_q^{d+1^-})}{\theta_i(\kappa_p^d, \kappa_q^{d+1})/|\theta_i(\kappa_p^d, \kappa_q^{d+1})|} \tag{1}$$

where $i$ represents the calculation under data $X_i$, $\kappa^+$ and $\kappa^-$ respectively represent the positive and negative value regions in the extracted information, $\theta$ represent the cosine similarity, and $KL$ represents the calculation of KL divergence between two representative information.

### 3.2 INNERSIGHTNET: AGGREGATION

The community aggregation algorithm is predicated on following foundational **principles**: Newman (2006) argue that using probabilistic mixed models and expectation maximization algorithm can detect a wide range of structural types without prior knowledge.

**Preliminaries:** Let's establish some foundational concepts firstly.

*Definition 1:* A **connection weight** between neurons can be categorized as an **activation connection** if its value is positive, or an **inhibition connection** if its value is negative.

*Definition 2:* A connection is considered **valid** if its activation weight exceeds a certain threshold $\xi_1$, or if its inhibition weight is below a certain threshold $\xi_2$. We denote valid connections by the value 1. Conversely, a connection is considered **invalid** if its activation weight is less than $\xi_3$, or its inhibition weight is greater than $\xi_4$. Invalid connections are denoted by the value 0.

Let $A_{i,k}^{d-1,d}$ ($I_{i,k}^{d-1,d}$) denote the activation (inhibition) connection between the $k$-th neuron in layer $d$ and the $i$-th neuron in layer $d-1$. Similarly, let $A_{k,j}^{d,d+1}$ ($I_{k,j}^{d,d+1}$) denote the activation (inhibition) connection between the $k$-th neuron in layer $d$ and the $j$-th neuron in layer $d+1$. Therefore, we define four connection matrices: $A^{d-1,d}$, $I^{d-1,d}$, $A^{d,d+1}$, and $I^{d,d+1}$. For clarity, these are denoted as $A$, $I$, $A^{'}$, and $I^{'}$, respectively.

*Definition 3:* The **probability of connection** refers to the likelihood that a specific connection between neurons is valid. We define $\tau_{i,k}^A$ and $\tau_{i,k}^I$ as the probabilities of the activation and inhibition connections between the $k$-th neuron in layer $d$ and the $i$-th neuron in layer $d-1$ being valid, respectively. Similarly, $\tau_{k,j}^{A^{'}}$ and $\tau_{k,j}^{I^{'}}$ denote the probabilities of the activation and inhibition connections between the $k$-th neuron in layer $d$ and the $j$-th neuron in layer $d+1$ being valid.

*Definition 4:* We define $\pi_c$ as the **probability** that a neuron belongs to the $c$-th community within the network, where $\sum \pi_c = 1$.

$\tau_{i,k}^A$ and $\tau_{i,k}^I$ as the probabilities of the activation and inhibition connections between the $k$-th neuron in layer $d$ and the $i$-th neuron in layer $d-1$. $\tau_{k,j}^{A^{'}}$ and $\tau_{k,j}^{I^{'}}$ are the probabilities of the activation and inhibition connections between the $k$-th neuron in layer $d$ and the $j$-th neuron in layer $d+1$.

Newman (2006) pointed out that the standard framework for fitting such models to a given dataset is likelihood maximization. To address this issues, we enter $g = \{g_k\}$ for calculating the expected log likelihood estimation. $g_k$ represents the community assignment of the $k$-th neuron in the $d$-th layer. The probabilities of $A$, $I$, $A^{'}$, $I^{'}$ and $g$ can be expressed by Eq. 2.

$$\mathbf{Pr}(A,I,A^{'},I^{'},g|\pi,\tau^A,\tau^I,\tau^{A^{'}},\tau^{I^{'}}) = \mathbf{Pr}(A,I,A^{'},I^{'},g,\pi|\tau^A,\tau^I,\tau^{A^{'}},\tau^{I^{'}})\cdot$$
$$\mathbf{Pr}(g|\pi,\tau^A,\tau^I,\tau^{A^{'}},\tau^{I^{'}}) \tag{2}$$

where the two factors in Eq.2 is shown in Eq.3 and Eq.4.

$$\mathbf{Pr}(A,I,A^{'},I^{'},g,\pi|\tau^A,\tau^I,\tau^{A^{'}},\tau^{I^{'}}) =$$
$$\prod_k\{\prod_i(\tau_{i,g_k}^A)^{A_{i,k}}(1-\tau_{i,g_k}^A)^{1-A_{i,k}}(\tau_{i,g_k}^I)^{I_{i,k}}(1-\tau_{i,g_k}^I)^{1-I_{i,k}}\}$$
$$\prod_k\{\prod_j(\tau_{g_k,j}^{A^{'}})^{A_{k,j}^{'}}(1-\tau_{g_k,j}^{A^{'}})^{1-A_{k,j}^{'}}(\tau_{g_k,j}^{I^{'}})^{I_{i,k}}(1-\tau_{g_k,j}^{I^{'}})^{1-I_{k,j}^{'}}\} \tag{3}$$

$$\mathbf{Pr}(g|\pi,\tau^A,\tau^I,\tau^{A^{'}},\tau^{I^{'}}) = \prod_k \pi_{g_k} \tag{4}$$

The expected log likelihood estimation $\mathcal{L}_g$ on $g = \{g_k\}$ can be obtained by Eq. 5.

$$\mathcal{L}_g = \sum_g \mathbf{Pr}(g|\pi,\tau^A,\tau^I,\tau^{A^{'}},\tau^{I^{'}}) \cdot \frac{1}{l_d} \ln \mathbf{Pr}(A,I,A^{'},I^{'}) \tag{5}$$

$q_{k,c}$ represents the probability of that the $k$-th neuron is assigned to $c$-th community. $l_d$ is the number of neurons in d-th layer. According to Bayesian, we can conclude that $q_{k,c}$ is represented as Eq. 6.

$$q_{k,c} = \frac{\mathbf{Pr}(A,I,A^{'},I^{'},g_k=c|\pi,\tau^A,\tau^I,\tau^{A^{'}},\tau^{I^{'}})}{\mathbf{Pr}(A,I,A^{'},I^{'}|\pi,\tau^A,\tau^I,\tau^{A^{'}},\tau^{I^{'}})} \tag{6}$$

Through Eq. 2 to Eq. 6, we maximize $\mathcal{L}_g$ to solve for the best assignment of neurons. Using the Lagrangian multiplier method, we can get $q_{k,c}$, $\pi_c$, $\tau_{i,k}^A$, $\tau_{i,k}^I$, $\tau_{k,j}^{A^{'}}$ and $\tau_{k,j}^{I^{'}}$ as Eq. 7.

$$q_{k,c} = \frac{p_{k,c}}{\sum_s p_{k,c}}, \pi_c = \frac{\sum_k q_{k,c}}{l_d}, \tau^A = \frac{\sum_k A_{i,k} q_{k,c}}{\sum_k q_{k,c}}$$

$$\tau^I = \frac{\sum_k I_{i,k} q_{k,c}}{\sum_k q_{k,c}}, \tau^{A'} = \frac{\sum_k A'_{k,j} q_{k,c}}{\sum_k q_{k,c}}, \tau^{I'} = \frac{\sum_k I'_{k,j} q_{k,c}}{\sum_k q_{k,c}} \tag{7}$$

where $p_{k,c}$ is shown as Eq. 8, $\varphi$ is equal to $c$ or $s$.

$$p_{k,\varphi} = \pi_c \cdot [\prod_i (\tau^A_{i,\varphi})^{A_{i,\varphi}} (1 - \tau^A_{i,\varphi})^{1-A_{i,k}} (\tau^I_{i,\varphi})^{I_{i,k}} (1 - \tau^I_{i,\varphi})^{1-I_{i,k}}] \cdot$$
$$[\prod_j (\tau^{A'}_{\varphi,j})^{A'_{\varphi,j}} (1 - \tau^{A'}_{\varphi,j})^{1-A'_{i,k}} (\tau^{I'}_{i,\varphi})^{I'_{k,j}} (1 - \tau^{I'}_{i\varphi,j})^{1-I'_{i,k}}] \tag{8}$$

See **appendix A.3** for detailed inferential proof process.

### 3.3 INNERSIGHTNET: EVALUATION

To confirm the best number of communities, we introduce $Q$. $Q$ determines whether the division is the most reasonable by measuring the consistency of input-output and structure-related metric.

**Similarity based on input sensitivity:** Inspired by Lange et al. (2022), we measure the similarity in input sensitivity of individual neurons in a community. Let $J^d_{x_i}$ be the $n \times m$ Jacobian matrix, where $x_i$ is the input, and $d$ is the layer index. The $i$-th row and $j$-th column in the $J^d_{x_i} \cdot J^{d}_{x_i}{}^T \in \mathbb{R}^{n \times n}$ is the similarity measure of sensitivity between the $i$-th and $j$-th neurons in layer $d$ when the input sample is $x_i$. The sensitivity similarity between the $i$-th and $j$-th neurons is:

$$S_{in}(i,j) = \frac{1}{K} |\sum_K^{k=1} J^d_{x_i} \cdot J^{d}_{x_i}{}^T|_{(i,j)} \tag{9}$$

where $K$ is the sample number of test set. The similarity of neurons within a community can be recorded as: $S_{in} = \sum S_{in}(i,j)/n(n-1), \forall i,j \in c, i > j$ and $i \neq j$.

**Similarity based on output representation:** To measure the consistency of outputs within a community, we introduce a new statistic called normalization consistency score (ncs). Firstly, we calculate the average output representation $\bar{F}_{out}$ of each neuron. Then, ncs calculates the output $F_{out}$ of each sample after input into the deep neural network, and get the standard deviation $s$.

$$s = \frac{1}{K-1} \sum_{k=1}^K \mathbf{norm}(F_{out} - \bar{F}_{out})^2; \; \text{ncs} = \frac{1}{s+1} \tag{10}$$

where **norm** is a normalization process, in order to ensure that ncs only measures the consistency, and is not affected by the values. $1/(s+1)$ converts the ncs to a consistency score between 0 and 1. The sign consistency metric $\Gamma$ is introduced to consider the directionality of numerical deviations.

$$\Gamma = \frac{\sum |F_{out} - \bar{F}_{out}| \cdot sign(F_{out} - \bar{F}_{out})}{\sum |F_{out} - \bar{F}_{out}|} \tag{11}$$

where $sign$ is the sign function. $S_{out} = |\Gamma| \cdot \text{ncs}$.

**Score based on Structure:** $S_{stru}$, a modularity variant, evaluates the consistency of neuronal behavior within a community and differences between communities. It generates four connection matrices using varying weight thresholds. A function $u(c_i, c_j)$ calculates the number of shared connections between $c_i$ and $c_j$, measuring intra-community similarity (when $i = j$) and inter-community differences (when $i \neq j$). Subsequently, we iterate over each pair of communities $(c_i, c_j)$, calculate and construct two connection matrices $U_{act}$ and $U_{inh}$, which the size is $i \times i$. Each element in the matrix represents the common connection between the corresponding communities. $S_{stru}$ needs to represent the activation and inhibition connections of weights, $S_{stru} = S^{act}_{stru} + S^{inh}_{stru}$, where the factors are calculated as shown in Eq. 12.

$$S_{stru}^{act/inh} = \sum_{i=1}^{n}(\frac{U_{act/inh}(c_i, c_j)}{\sum U_{act/inh} + \zeta} - (\frac{\sum_j U_{act/inh}(c_i, c_j)}{\sum U_{act/inh} + \zeta})^2) \tag{12}$$

where $n$ is the number of communities, $\sum U_{act/inh}$ is the sum of all elements. $\zeta = 10^{-6}$ to avoid the denominator to 0. The evaluation indicator $Q$ can be calculated by weighting $S_{in}$, $S_{out}$ and $S_{stru}$.

$$Q = \sum_{c_i=1}^{c}\omega_{c_i} \cdot (\alpha S_{in}^{c_i} + S_{in}^{c_i}) + \beta S_{stru} \tag{13}$$

where $\omega_{c_i} = \text{len}(c_i)/\sum\text{len}(c_i)$ is the weight. $\text{len}(c_i)$ represents the number of neurons in $c_i$. $\alpha$ and $\beta$ are the hyper-parameters to trade-off the $S_{in}$, $S_{out}$ and $S_{stru}$. We demonstrate the three steps of the InnerSightNet: 'neuronization', 'aggregation', and 'evaluation', as shown in Algorithm 1.

---

**Algorithm 1** Inner Information Analysis Algorithm for Deep Neural Network based on Community

---

1: **Input:** deep neural network $N$
2: **Output:** optimal neuron community partition
3: **if** $N$ is a convolutional layer **then**
4:     Neuronize($N$)                                                 ▷ According to Eq. 1
5: **end if**
6: Initialize optimal evaluation metric $Q_{\text{opt}} \leftarrow -\infty$
7: Initialize optimal community count $r_{\text{opt}} \leftarrow 0$
8: **for** $r \leftarrow 1$ **to** $20$ **do**
9:     **for** iter $\leftarrow 1$ **to** $200$ **do**
10:         Perform EM algorithm to update community partition
11:         Randomly initialize model parameters $\pi_c, \tau_{ci}^{act}, \tau_{ci}^{inh}, \tau_{cj'}^{act}, \tau_{cj'}^{inh}$    ▷ According to Eq. 7
12:         Initialize probability matrix $q_{kc}(k, c)$                          ▷ According to Eq. 7
13:         Compute probability $q_{kc}(k, c)$
14:         Update model parameters based on $q_{kc}(k, c)$
15:     **end for**
16:     $Q \leftarrow$ Calculate evaluation metric $Q$ value                  ▷ According to Eq. 13
17:     **if** $Q > Q_{\text{opt}}$ **then**
18:         $Q_{\text{opt}} \leftarrow Q$
19:         $r_{\text{opt}} \leftarrow r$
20:         $q_{kc_{\text{opt}}} \leftarrow q_{kc}$
21:     **end if**
22: **end for**
23: **return** community partition for $r_{\text{opt}}, q_{kc_{\text{opt}}}$

---

## 4 EXPERIMENTS & APPLICATIONS

We use differential output analysis and perturbation statistical analysis to analyze the results of InnerSightNet. For more details, please refer to **appendix A.5**.

### 4.1 INNERSIGHTNET IN LINEAR LAYER AND CONVOLUTIONAL LAYER

We use InnerSightNet to analyse two typical structures: linear layer and convolutional layer. We use differential output analysis and perturbation statistical analysis (an improved method based on Watanabe et al. (2018)) to explore the representation of community.

#### 4.1.1 INNERSIGHTNET IN LINEAR LAYER

**Overview:** During our investigation into the InnerSightNet on linear layers, we focus on a classic benchmark: the MNIST recognition task, a problem of classifying into 10 categories. We design a linear neural network Rosenblatt (1958) with three hidden layers, with 128, 64, and 32 hidden nodes respectively, and the output layer is a $1 \times 10$ vector . The MNIST dataset LeCun et al. (1998)

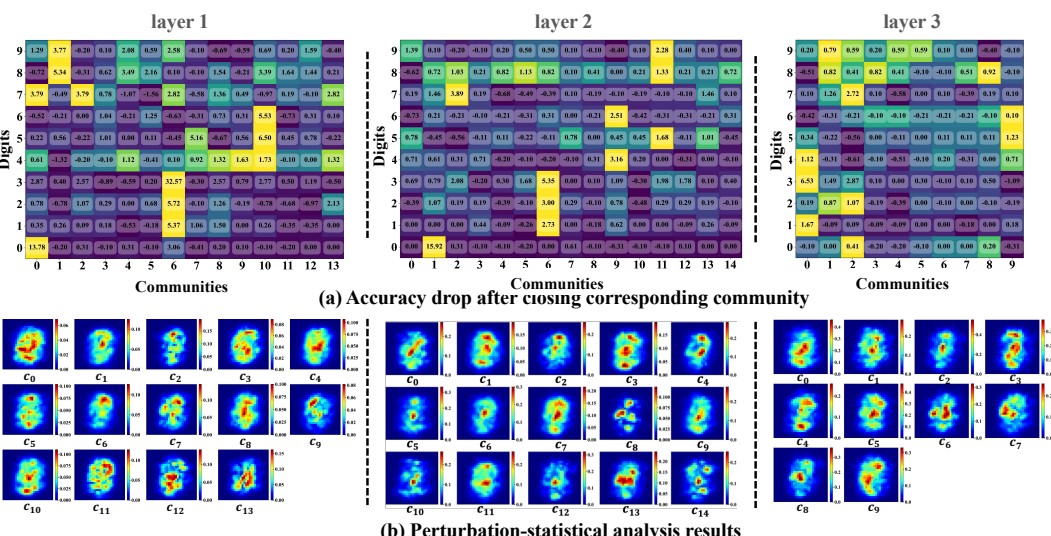

Figure 1: (a) Accuracy drop after closing corresponding community. (b) Perturbation-statistical analysis. Note: to clearly show the attention of each community, we do not normalize the colorbar.

consists of handwritten digits, comprised of 60,000 samples within the train-set and an additional 10,000 samples in the test-set. Each sample takes the form of a 28 × 28 pixel grayscale image, where each pixel's value, ranging from 0 to 255, signifies varying levels of color intensity.

We conduct InnerSightNet on the well-trained linear neural network. Due to the hidden nodes naturally play the role of neurons, we directly adopt the 'Aggregation' to identify various communities. We determine the ideal number of communities for the three hidden layers of the network as 14, 15, and 10 based on $Q$ value. To clarify the specific impact of each community on output accuracy, we use differential output analysis. Specifically, we shut down all neurons in the community at once and record the decrease in accuracy, as shown in Fig. 1 (a). We also use perturbation-statistical analysis to visualize the sensitivity of each community to input, as shown in Fig. 1 (b).

**Community function analysis and finding key communities:** According to Fig. 1 (a), we observe $c_6$ in the first layer is crucial for identifying digit 3. When it is removed (without re-training), the accuracy decreases by 32.57%, and the impacts on digit 1 and 2 are significant, with accuracy decreases by 5.37% and 5.72%. This indicates $c_6$ may be responsible for extracting certain shared geometric features, such as curves and angles. $c_1$ in the second layer is crucial for recognizing digit 0, with an accuracy decrease of 15.92%. $c_1$ captures global closed shapes such as circles. From 1 (b), it can be seen that $c_1$ in second layer is more sensitive to the edges of such structures.

In terms of specificity analysis, in the second layer, we see that $c_6$ has a significant impact on identifying digit 1, 2 and 3 (with a decrease of 2.73%, 3.00%, and 5.35%). This indicates $c_6$ can capture the vertical or diagonal features of these digits. The $c_2$ of the second layer has a significant impact on digit 7 and 8, with a decrease of 3.89% and 1.03%. This may indicate $c_2$ has a high sensitivity to the combined shape of vertical lines and curves. The $c_{10}$ in the first layer has the greatest impact on digit 5 with an accuracy decrease of 7.6%, and the impact on digit 6 is second, with an accuracy decrease of 5.53%. The accuracy of digit 8 has decreased by 3.39%, and the accuracy of the digit 3 has decreased by 2.77%. This indicates that $c_{10}$ may have captured common specific features of the digit 5, 6, 8 and 3, such as the curve in the lower right area. From the perturbation-statistical analysis in Fig. Fig. 1 (b), it can be seen that $c_{10}$ in the first layer is more sensitive to the curve in the lower right region. In the second layer, $c_6$ has a significant impact on the digit 1, 2, and 3. After closing the community, the accuracy decreases by 2.73%, 3.00%, and 5.35%. This indicates that $c_6$ captures the common feature of digit 1, 2, and 3, which is the vertical line segment (digit 1 is entirely composed of vertical lines, the top and bottom of digit 2 are usually connected by a vertical line, and the upper and lower arcs of digit 3 visually form an implicit connection through the vertical symmetry in the middle). From the perturbation statistical analysis

in Fig. 1 (b), it can be seen that $c_6$ is more sensitive to vertical line segments. In terms of redundancy analysis, for digit 5, the impact of $c_{10}$ and $c_7$ in the first layer is significant (decreased by 6.50% and 5.16%). This may indicate these two communities capture the curve or combination features from different perspectives, and there may be redundant feature extraction to improve the network's fault tolerance.

### 4.1.2 INNERSIGHTNET IN CONVOLUTIONAL LAYER

**Overview:** During investigating the InnerSightNet within convolutional neural networks LeCun et al. (1989), we select the task of cat and dog classification. We design a network architecture that consists of three convolutional layers (consisting of 32, 64, and 128 kernels). This is followed by three linear layers for binary classification. The AFHQ dataset Choi et al. (2020) has been chosen. Specifically, the train-set consists of 5,153 images of cats and 4,739 images of dogs, while the test-set includes 500 images from each category. Through the InnerSightNet, we perform community detection on the well-trained convolutional layer. Based on the $Q$-value, we determine that the most ideal number of communities for the three convolutional layers in a convolutional layer is 3, 3, and 4. We visualize the sensitivity of each community to input data using perturbation-statistical analysis, as shown in Fig. 2 (a). According to the perturbation-statistical analysis, $c_1$ and $c_2$ in the first convolutional layer, $c_1$ and $c_2$ in the second convolutional layer, $c_0$, $c_2$, and $c_3$ in the third convolutional layer are defined as key communities for this task, as shown in Fig. 2 (b).

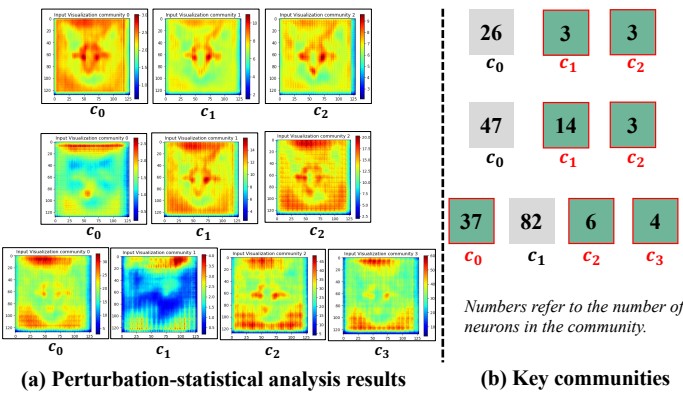

**(a) Perturbation-statistical analysis results**

**(b) Key communities**

*Numbers refer to the number of neurons in the community.*

Table 1: Acc after closing corresponding community.

| | layer 1 | | | Acc (%) |
|---|---|---|---|---|
| $c_0$ | $c_1$ | $c_2$ | $c_3$ | |
| √ | | | ○ | 98.5 |
| | √ | | ○ | 98.1 |
| | | √ | ○ | 97.9 |
| √ | √ | | ○ | 95.8 |
| | layer 2 | | | Acc (%) |
| √ | | | ○ | 98.7 |
| | √ | | ○ | 94.2 |
| | | √ | ○ | 98.1 |
| √ | √ | | ○ | 69.9 |
| | layer 3 | | | Acc (%) |
| √ | | | | 96.9 |
| | √ | | | 97.5 |
| | | √ | | **98.8(+)** |
| | | | √ | 97.7 |
| √ | | √ | √ | 55.5 |

Figure 2: (a) Perturbation-statistical analysis results. (b) Key communities. The green communities representing key neurons and gray representing non-key neurons. Numbers refer to the number of neurons in the community.

**Community function analysis and finding key communities:** To further investigate the role of these key communities, we close each community one by one (i.e., setting the convolutional kernel output of the corresponding index within the community to 0), and record the impact on the accuracy. The results are shown in Table 1. If all the communities are in an open state, the accuracy is 98.7%. In Table 1, the '√' indicates that the community is closed, and the '○' indicates that there is no $c_3$ community in the first and second convolutional layers.

According to table 1, in the first convolutional layer, the $c_0$ community contains relatively less information, which has little impact on the accuracy. In contrast, the $c_1$ and $c_2$ communities contain more information. When both the $c_1$ and $c_2$ communities are closed simultaneously, the accuracy decreases significantly, indicating that the information in the $c_1$ and $c_2$ communities has a certain degree of complementarity in recognition. The same logic also applies to the 2-nd layer. In the 3-rd layer, the $c_1$ community contains less recognition feature information compared to others, while the information in the $c_0$, $c_2$, and $c_3$ communities together form a complementary recognition feature.

In addition, we observed that when $c_1$ in the third layer is closed, the accuracy is actually improved. Based on the perturbation statistical analysis results, as shown in 2 (a), we can confirm that $c_1$ mainly contains features unrelated to the recognition task (which we define as 'noise'). Similarly, from 2

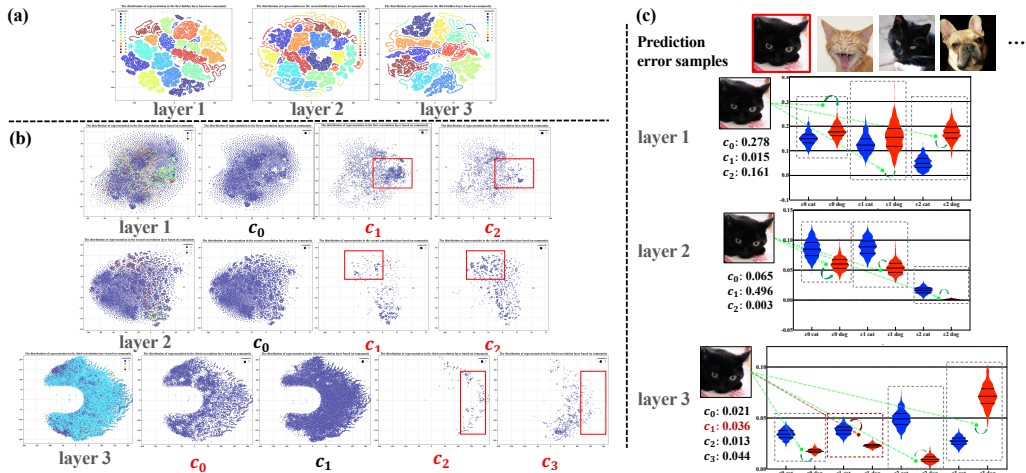

Figure 3: (a) The representation distribution in the linear layers. (b) The representation distribution in the convolutional layers. (c) The distribution of community activation levels in convolutional layers and the analysis of the error prediction sample.

(a), we can identify that the $c_0$ in the first layer and the $c_0$ in the second layer are both 'noise'. During the training process of the model, the convolutional layers inevitably fit some noisy data. Closing these neurons during the testing phase reduces overfitting and enhances robustness. This processing makes the model more accurate in capturing core features to improve the generalization ability.

From Fig. 2 (a), it can be seen that the convolution kernels located within the key community mainly focus on the key feature areas for cat and dog classification, while those in non-key communities focus on the non recognition feature areas or have insufficient attention to the recognition features. We attempt to close the non-key communities of the three convolutional layers ($c_0$ in the first layer, $c_0$ in the second layer, and $c_1$ in the third layer). The result shows that although the accuracy of cat and dog classification decreased to 93.6% (a decrease of 5.1%), we achieve a high recognition accuracy with only 70 neurons. Compared with 224 neurons using the entire convolutional layers, the number of neurons used decreased by 68.75%, providing a new perspective for network pruning.

## 4.2 VISUALIZING THE REPRESENTATIONS OF NEURONS WITHIN THE COMMUNITY

To more intuitively reveal the inherent consistency of neuron representations within the same community, we visualize the representations of neurons. We traverse the test-set and calculate the average representation of each neuron as a benchmark. We fed the samples from test-set one by one and record the output values of each neuron. For neurons within a specific community, we use the output value corresponding to their index as their representation. For neurons that do not belong to the community, we use the average as their representation. We collect representation data of specific community neurons and record the community index to which these representations belong. We use T-SNE to reduce the dimensionality of these representations, as shown in Fig. 3 (a) and (b).

From Fig. 3(a), neurons within the same community cluster due to similar functions and response patterns, reflecting shared feature preferences formed during training. Fig. 3(b) highlights complementarity and differences in key communities. In the first convolutional layer, $c_1$ and $c_2$ share similarities in feature distribution, explaining their minor individual impact on accuracy but a significant drop when both are closed. Their distinct feature differences, marked in red boxes, justify their assignment to separate communities.

## 4.3 COMMUNITY ACTIVATION LEVEL AND ANALYSIS OF ERROR PREDICTION SAMPLES

To accurately evaluate the activity level of neurons within the community, we adopt the following statistical measurement. We define 'community activation level': it refers to the average level of activation values of all neurons within the community. We record the activation outputs of each

community in each convolutional layer when the input image is a cat. Similarly, we also record the situation when the input image is a dog. We quantitatively describe the distribution of community activation levels using violin plots, as shown in Fig. 3 (c). In cat and dog recognition task, we focus on those error prediction samples and analyze the community activation levels, aiming to explore the reasons behind classification errors. In Fig. 3 (c), we present some prediction errors. Taking the first image as a case, we demonstrate the activation level of each community in convolutional layer when the misclassified sample is fed. The activation values in the first layer suggest that the image tends to activate patterns associated with dogs, which is consistent with observations in the second layer. However, in the third layer, the community activation level of $c_1$ is closer to the distribution of cat, while in $c_0$, $c_2$, and $c_3$ are again biased towards dogs. Importantly, $c_0$, $c_2$, and $c_3$ in the third layer are the key communities responsible for key identification, while $c_1$ is the non-key community.

## 4.4 COMPARE TO OTHER METHODS

**Searching for Noise Neuron Communities:** In Table 2, we demonstrate that turning off noisy neurons in the last layer can improve performance. This is because the noise neuron community focuses more on non-task features, and deleting these neurons makes the networks pay more attention to features. To demonstrate the effectiveness of InnerSightNet in searching for noisy neuron communities, we compare InnerSightNet with Filan et al. (2021), Hod et al. (2021), and Liu et al. (2023) to search for noisy neurons in the last layer and test the improvement in final accuracy. Filan *et al.* investigate the concept of 'clusterability', focusing on dividing neurons into groups with strong internal connectivity and weak external connectivity. Hod *et al.* focus on quantifying the local specialization of neural networks, where clusters of neurons are linked to comprehensible sub-tasks. Liu *et al.* propose Brain-Inspired Modular Training, which enhances network modularity and interpretability by embedding neurons in a geometric space, penalizing connection lengths during training. We choose MNIST and AFHQ as datasets, and select linear and convolutional layers to be tested, respectively.

Table 2: The results of searching for noise neuron communities.

|  | MNIST | AFHQ |
|---|---|---|
| Filan *et al.* | 92.58%±0.062% | 98.56%±0.135% |
| Hod *et al.* | 92.61%±0.014% | 98.60%±0.107% |
| Liu *et al.* | 92.63%±0.020% | 98.68%±0.075% |
| InnerSightNet | 92.69%±0.008% | 98.78%±0.033% |

Table 3: The results of network pruning based on key neurons.

|  | num | Acc |
|---|---|---|
| Filan *et al.* | 105 | 94.2% |
| Hod *et al.* | 81 | 93.5% |
| Liu *et al.* | 75 | 93.4% |
| InnerSightNet | 70 | 93.6% |

**Network Pruning Based on Core Neuron Community:** To verify the superiority of InnerSightNet in locating key neurons, we choose the convolutional layers trained on AFHQ as the network to be tested. Meanwhile, using Filan et al. (2021), Hod et al. (2021), and Liu et al. (2023) as baseline methods to search for the key neurons in the neural network. From Table 3, it can be seen that InnerSightNet has significant advantages in the search of key neurons. Although the accuracy of the Filan et al. (2021) method is 0.6% higher than that of InnerSightNet, it uses 15.625% more neurons than InnerSightNet. Overall, InnerSightNet performs better in searching for key neurons. This is mainly due to the fact that InnerSightNet considers the connection strength and probability between different neurons and layers, rather than focuses not only on a single layer or individual neuron. InnerSightNet is not only suitable for network pruning, but can also be applied to other fields.

## 5 CONCLUSION AND FUTURE WORK

**In this work**, we use the inherent characteristics of neurons in learnable units to partition neurons into communities. InnerSightNet adaptively searches for the best number of communities and displays the sensitive areas of concern to the community based on roles and functionalities analysis. We analyze the inference process of neural networks from the community perspective, avoiding the limitations of only analyzing single layers or individual neurons. **Many future works follow.** According to our algorithm, community-based analysis methods can be potentially applied to the analysis of other tasks, such as analyzing the flow of abstract concepts during image generation from generative networks, dynamic problems during network training, etc. Our future work also is based on community analysis to improve our understanding in deep neural networks.

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

## A APPENDIX

### A.1 APPENDIX ABSTRACT

In this work, InnerSightNet can be divided into three primary phases: Initially, it executes a process of 'neuronization', transforming learnable units into a structured network of neurons. Subsequently, these neurons are clustered into distinct communities according to representation attributes. The final stage involves the examination of these communities' roles and functionalities to make sure the best community partitioning. In additional, we use differential output analysis and perturbation-statistical method to unpick the neural tapestry of decision-making. In the appendix, we elaborate on the theoretical background (A.2) and supplement the detailed derivation process of the formulas (A.3) cited in the main text. In addition, we also provide specific details of the algorithm implementation (A.4), methods of roles and functionalities analysis in InnerSightNet (A.5), attach extensive experimental results (A.6) and limitations (A.7).

### A.2 THEORETICAL BACKGROUND

In analyzing complex data structures, especially those containing unobserved or implicit variables, probability models demonstrate their powerful ability to effectively reveal the hidden structures behind the data. This type of model introduces probability distribution to describe the process of data generation, which can not only handle inherent uncertainty properly, but also use statistical methods

to accurately estimate model parameters. The core advantage of probability models lies in their ability to use parameterization to characterize the interdependence between variables. Especially in classification or clustering problems, the Expectation Maximization (EM) algorithm is often used to infer the parameters of these probability models. This algorithm optimizes parameter estimation through an iterative process to adapt to the statistical characteristics of observed data.

***Theorem: EM parameter estimation for probabilistic models:*** Given a set of observation data $A^{d-1,d}$, $I^{d-1,d}$, $A^{d,d+1}$ and $I^{d,d+1}$ (To facility a cleaner description, we use $A$, $I$, $A'$, and $I'$ install.), we consider a probability model that takes into account the parameters $\{\pi_c, \tau_{i,k}^A, \tau_{i,k}^I, \tau_{k,j}^{A'}, \tau_{k,j}^{I'}\}$ describe the process of result generation. The goal of the model is to maximize the likelihood function of the observed data, which typically involves marginalization of implicit variables. In this context, we can describe the estimation method of model parameters through the following theorem:

Let $\pi_c$ represents the prior probability of class $c$, $\{\tau_{i,k}^A, \tau_{i,k}^I, \tau_{k,j}^{A'}, \tau_{k,j}^{I'}\}|_{k=c}$ represent the conditional probability of a given class $c$, respectively. For each observation $k$ and class $c$, define $q_{k,c}$, where is the posterior probability that observation $k$ belongs to class $c$. The parameters $\{\pi_c, \tau_{i,k}^A, \tau_{i,k}^I, \tau_{k,j}^{A'}, \tau_{k,j}^{I'}\}$ can be iteratively estimated through the following expectation maximization steps:

**(Step E)** Estimate the posterior probability based on the current parameters:

$$q_{k,c} = p_{k,c}/\sum_s p_{k,c}, \tag{14}$$

where $p_{k,c}$ is shown as Eq. 15, $\varphi$ is equal to $c$ or $s$.

$$p_{k,\varphi} = \pi_c \cdot [\prod_i (\tau_{i,\varphi}^A)^{A_{i,\varphi}} (1 - \tau_{i,\varphi}^A)^{1-A_{i,k}} (\tau_{i,\varphi}^I)^{I_{i,k}} (1 - \tau_{i,\varphi}^I)^{1-I_{i,k}}] \cdot$$
$$[\prod_j (\tau_{\varphi,j}^{A'})^{A'_{\varphi,j}} (1 - \tau_{\varphi,j}^{A'})^{1-A'_{i,k}} (\tau_{i,\varphi}^{I'})^{I'_{k,j}} (1 - \tau_{i\varphi,j}^{I'})^{1-I'_{i,k}}] \tag{15}$$

**(Step M)** Update the model parameters to maximize the likelihood function of the observed data:

$$\pi_c = \frac{\sum_k q_{k,c}}{l_d}, \tau^A = \frac{\sum_k A_{i,k} q_{k,c}}{\sum_k q_{k,c}}, \tau^I = \frac{\sum_k I_{i,k} q_{k,c}}{\sum_k q_{k,c}}$$
$$\tau^{A'} = \frac{\sum_k A'_{k,j} q_{k,c}}{\sum_k q_{k,c}}, \tau^{I'} = \frac{\sum_k I'_{k,j} q_{k,c}}{\sum_k q_{k,c}} \tag{16}$$

By adopting this theorem, we explain how to use the expectation maximization algorithm to estimate the parameters of a probability model under known observation data conditions. This method provides the mathematical foundation for revealing the hidden category structure in the data.

When applying this theorem for parameter estimation, we initially determine the probability of each data belonging to different classes through **Step E**, that is, implementing 'soft clustering'. Subsequently, in **Step M**, we adjust the model parameters to enhance the overall likelihood of these probability distributions. Through repeated iterations, the algorithm will eventually converge and obtain the optimal estimate of parameters, thereby revealing the implicit structure within the data.

A.3 PROOF OF SECTION 2.2

The description of Eq. 2 to Eq. 8 in main text is the basis of the **InnerSightNet: aggregation**. Here, we provide the detailed derivation process for Eq. 2 to Eq. 8.

According to Bayesian theorem:

$$\mathbf{Pr}(A, I, A', I', g|\pi, \tau^A, \tau^I, \tau^{A'}, \tau^{I'}) = \mathbf{Pr}(A, I, A', I', g, \pi|\tau^A, \tau^I, \tau^{A'}, \tau^{I'}) \cdot$$
$$\mathbf{Pr}(g|\pi, \tau^A, \tau^I, \tau^{A'}, \tau^{I'}) \tag{17}$$

Based on the connection matrix and joint probability, we can calculate the two factors in Eq. 17 separately. For factor $\mathbf{Pr}(A, I, A', I', g, \pi | \tau^A, \tau^I, \tau^{A'}, \tau^{I'})$, where $\{\tau_{i,k}^A, \tau_{i,k}^I, \tau_{k,j}^{A'}, \tau_{k,j}^{I'}\}$ is used as the condition, calculate the probability distribution of $\mathbf{Pr}(A, I, A', I', g, \pi)$:

$$
\begin{aligned}
&\mathbf{Pr}(A, I, A', I', g, \pi | \tau^A, \tau^I, \tau^{A'}, \tau^{I'}) = \\
&\prod_k \{\prod_i (\tau_{i,g_k}^A)^{A_{i,k}} (1 - \tau_{i,g_k}^A)^{1-A_{i,k}} (\tau_{i,g_k}^I)^{I_{i,k}} (1 - \tau_{i,g_k}^I)^{1-I_{i,k}}\} \\
&\prod_k \{\prod_j (\tau_{g_k,j}^{A'})^{A'_{k,j}} (1 - \tau_{g_k,j}^{A'})^{1-A'_{k,j}} (\tau_{g_k,j}^{I'})^{I'_{i,k}} (1 - \tau_{g_k,j}^{I'})^{1-I'_{k,j}}\}
\end{aligned}
\tag{18}
$$

For factor $\mathbf{Pr}(g | \pi, \tau^A, \tau^I, \tau^{A'}, \tau^{I'})$, where $\{\pi, \tau_{i,k}^A, \tau_{i,k}^I, \tau_{k,j}^{A'}, \tau_{k,j}^{I'}\}$ is used as the condition, calculate the probability distribution of $\mathbf{Pr}(g)$

$$
\mathbf{Pr}(g | \pi, \tau^A, \tau^I, \tau^{A'}, \tau^{I'}) = \prod_k \pi_{g_k}
\tag{19}
$$

Due to the fact that the probability distribution of $\mathbf{Pr}(A, I, A', I', g, \pi | \tau^A, \tau^I, \tau^{A'}, \tau^{I'})$ is in the form of continuous multiplication, it is very friendly for logarithmic calculations. Therefore, the logarithmic likelihood function of the probability distribution is:

$$
\begin{aligned}
\mathcal{L} =& \frac{1}{l_d} \ln \mathbf{Pr}(A, I, A', I', g | \pi, \tau^A, \tau^I, \tau^{A'}, \tau^{I'}) = \\
& \frac{1}{l_d} \ln\{\mathbf{Pr}(A, I, A', I', g, \pi | \tau^A, \tau^I, \tau^{A'}, \tau^{I'}) \cdot \mathbf{Pr}(g | \pi, \tau^A, \tau^I, \tau^{A'}, \tau^{I'})\} \\
=& \frac{1}{l_d} \ln\{\mathbf{Pr}(A, I, A', I', g, \pi | \tau^A, \tau^I, \tau^{A'}, \tau^{I'})\} + \frac{1}{l_d} \ln\{\mathbf{Pr}(g | \pi, \tau^A, \tau^I, \tau^{A'}, \tau^{I'})\} \\
=& \frac{1}{l_d} \ln \prod_k \{\prod_i (\tau_{i,g_k}^A)^{A_{i,k}} (1 - \tau_{i,g_k}^A)^{1-A_{i,k}} (\tau_{i,g_k}^I)^{I_{i,k}} (1 - \tau_{i,g_k}^I)^{1-I_{i,k}}\} \\
& \prod_k \{\prod_j (\tau_{g_k,j}^{A'})^{A'_{k,j}} (1 - \tau_{g_k,j}^{A'})^{1-A'_{k,j}} (\tau_{g_k,j}^{I'})^{I'_{i,k}} (1 - \tau_{g_k,j}^{I'})^{1-I'_{k,j}}\} + \frac{1}{l_d} \ln\{\prod_k \pi_{g_k}\} \\
=& \frac{1}{l_d} \ln \sum_k \sum_i \{A_{i,k} \ln \tau_{i,g_k}^A + (1 - A_{i,k}) \ln(1 - \tau_{i,g_k}^A) + I_{i,k} \ln \tau_{i,g_k}^I + (1 - I_{i,k}) \\
& \ln(1 - \tau_{i,g_k}^I)\} + \sum_j \{A'_{k,j} \ln \tau_{g_k,j}^{A'} + (1 - A'_{k,j}) \ln(1 - \tau_{g_k,j}^{A'}) + I'_{i,k} \ln \tau_{g_k,j}^{I'} + \\
& (1 - I'_{k,j}) \ln(1 - \tau_{g_k,j}^{I'})\} + \ln\{\pi_{g_k}\}\}
\end{aligned}
\tag{20}
$$

where $l_d$ is the member of the neurons in $d$-th layer. Since the variable $g$ is unknown in Eq. 20, we calculate the expected value of the likelihood function on the implicit variable set $g = \{g_k\}$.

$$
\mathcal{L}_g = \sum_g \mathbf{Pr}(g | \pi, A, I, A', I', \tau^A, \tau^I, \tau^{A'}, \tau^{I'}) \cdot \mathcal{L}
\tag{21}
$$

Substitute Eq. 20 into Eq. 21, we can get,

$$
\begin{aligned}
\mathcal{L}_g =& \frac{1}{l_d} \sum_{k,c} q_{k,c} \{\ln \pi_c + \sum_i (A_{i,k} \ln \tau_{i,k}^A + (1 - A_{i,k}) \ln(1 - \tau_{i,k}^A)) + \sum_i (I_{i,k} \ln \tau_{i,k}^I + \\
& (1 - I_{i,k}) \ln(1 - \tau_{i,k}^I)) + \sum_j (A_{k,j} \ln \tau_{k,j}^{A'} + (1 - A'_{k,j}) \ln(1 - \tau_{k,j}^{A'})) + \sum_j (I_{k,j} \ln \tau_{k,j}^{I'} \\
& + (1 - I'_{k,j}) \ln(1 - \tau_{k,j}^{I'}))\}
\end{aligned}
\tag{22}
$$

$q_{k,c}$ represents the probability of that the $k$-th neuron is assigned to $c$-th community. According to Bayesian formula, we can conclude that $q_{k,c}$ is represented as Eq. 23.

$$q_{k,c} = \frac{\mathbf{Pr}(A, I, A^{'}, I^{'}, g_k = c | \pi, \tau^A, \tau^I, \tau^{A^{'}}, \tau^{I^{'}})}{\mathbf{Pr}(A, I, A^{'}, I^{'} | \pi, \tau^A, \tau^I, \tau^{A^{'}}, \tau^{I^{'}})} \tag{23}$$

For the numerator of $q_{k,c}$,

$$\mathbf{Pr}(A, I, A^{'}, I^{'}, g_k = c | \pi, \tau^A, \tau^I, \tau^{A^{'}}, \tau^{I^{'}})$$
$$= \{\sum_{g_1} \sum_{g_2} ... \sum_{g_k}\}|_{g_k=c} \mathbf{Pr}(A, I, A^{'}, I^{'}, g | \pi, \tau^A, \tau^I, \tau^{A^{'}}, \tau^{I^{'}}) \Rightarrow \text{donate as factor } \mathbf{B} \tag{24}$$
$$i.e., \ \mathbf{Pr}(A, I, A^{'}, I^{'}, g_k = c | \pi, \tau^A, \tau^I, \tau^{A^{'}}, \tau^{I^{'}}) = \text{factor } \mathbf{B}|_{g_k=c} \times \text{factor } \mathbf{B}|_{g_k \neq c}$$

where,

$$\text{(i) factor } \mathbf{B}|_{g_k=c} = \pi_c \cdot \prod_i (\tau_{i,c}^A)^{A_{i,c}} (1 - \tau_{i,c}^A)^{1-A_{i,c}} (\tau_{i,c}^I)^{I_{i,c}} (1 - \tau_{i,c}^I)^{1-I_{i,c}}.$$
$$\prod_j (\tau_{c,j}^{A^{'}})^{A_{c,j}^{'}} (1 - \tau_{c,j}^{A^{'}})^{1-A_{c,j}^{'}} (\tau_{c,j}^{I^{'}})^{I_{i,c}^{'}} (1 - \tau_{c,j}^{I^{'}})^{1-I_{c,j}^{'}}$$
$$\text{(ii) factor } \mathbf{B}|_{g_k \neq c} = \prod_{g_k \neq c} \sum_s \pi_s \cdot \prod_i (\tau_{i,s}^A)^{A_{i,s}} (1 - \tau_{i,s}^A)^{1-A_{i,s}} (\tau_{i,s}^I)^{I_{i,s}} (1 - \tau_{i,s}^I)^{1-I_{i,s}}. \tag{25}$$
$$\prod_j (\tau_{s,j}^{A^{'}})^{A_{s,j}^{'}} (1 - \tau_{s,j}^{A^{'}})^{1-A_{s,j}^{'}} (\tau_{s,j}^{I^{'}})^{I_{i,s}^{'}} (1 - \tau_{s,j}^{I^{'}})^{1-I_{s,j}^{'}}$$

For the denominator of $q_{k,c}$,

$$\mathbf{Pr}(A, I, A^{'}, I^{'} | \pi, \tau^A, \tau^I, \tau^{A^{'}}, \tau^{I^{'}}) = \{\sum_{g_1} ... \sum_{g_k}\} \mathbf{Pr}(A, I, A^{'}, I^{'}, g | \pi, \tau^A, \tau^I, \tau^{A^{'}}, \tau^{I^{'}})$$
$$= \prod_k \sum_s \pi_s \cdot \prod_i (\tau_{i,s}^A)^{A_{i,s}} (1 - \tau_{i,s}^A)^{1-A_{i,s}} (\tau_{i,s}^I)^{I_{i,s}} (1 - \tau_{i,s}^I)^{1-I_{i,s}}. \tag{26}$$
$$\prod_j (\tau_{s,j}^{A^{'}})^{A_{s,j}^{'}} (1 - \tau_{s,j}^{A^{'}})^{1-A_{s,j}^{'}} (\tau_{s,j}^{I^{'}})^{I_{i,s}^{'}} (1 - \tau_{s,j}^{I^{'}})^{1-I_{s,j}^{'}}$$

Therefore, we can get $q_{k,c}$ as follow,

$$q_{k,c} = \frac{p_{k,c}}{\sum_s p_{k,c}} \tag{27}$$

where $p_{k,c}$ is shown as Eq. 28, $\varphi$ is equal to $c$ or $s$.

$$p_{k,\varphi} = \pi_c \cdot [\prod_i (\tau_{i,\varphi}^A)^{A_{i,\varphi}} (1 - \tau_{i,\varphi}^A)^{1-A_{i,k}} (\tau_{i,\varphi}^I)^{I_{i,k}} (1 - \tau_{i,\varphi}^I)^{1-I_{i,k}}].$$
$$[\prod_j (\tau_{\varphi,j}^{A^{'}})^{A^{'}_{\varphi,j}} (1 - \tau_{\varphi,j}^{A^{'}})^{1-A_{i,k}^{'}} (\tau_{i,\varphi}^{I^{'}})^{I_{k,j}^{'}} (1 - \tau_{i\varphi,j}^{I^{'}})^{1-I_{i,k}^{'}}] \tag{28}$$

Currently, we have the likelihood function $\mathcal{L}_g$ and the constraint $\sum_c \pi_c = 1$. For this type of optimization problem with multiple variables and constraints, it can be transformed into a problem with a set of equations and can be solved through the Lagrange multiplier method. We define a new function $h$ as $h = mathbf\mathcal{L}_g - \alpha \sum_c \pi_c$, where $\alpha$ is a constant, and for function $h$, the best value exists if the following conditions are met.

$$\nabla_{\pi_c} h = 0, \nabla_{\tau_{i,c}^A} h = 0, \nabla_{\tau_{i,c}^I} h = 0, \nabla_{\tau_{c,j}^{A^{'}}} h = 0, \nabla_{\tau_{c,j}^{I^{'}}} h = 0 \tag{29}$$

where $\{\tau_{i,c}^A, \tau_{i,c}^I, \tau_{c,j}^{A'}, \tau_{c,j}^{I'}\}$ are the Lagrange multipliers.

For $\nabla_{\pi_c} h = 0$, we can get,

$$
\begin{aligned}
&\nabla_{\pi_c} (\mathcal{L}_g - \alpha\textstyle\sum_c \pi_c) = 0 \\
\Rightarrow &\nabla_{\pi_c} \mathcal{L}_g = \alpha \\
\Rightarrow &\nabla_{\pi_c} \frac{1}{l_d}\textstyle\sum_{k,c} q_{k,c}\{\ln \pi_c + \sum_i(A_{i,k}\ln \tau_{i,k}^A + (1-A_{i,k})\ln(1-\tau_{i,k}^A)) + \sum_i(I_{i,k}\ln \tau_{i,k}^I + \\
&(1-I_{i,k})\ln(1-\tau_{i,k}^I)) + \sum_j(A_{k,j}\ln \tau_{k,j}^{A'} + (1-A_{k,j}')\ln(1-\tau_{k,j}^{A'})) + \sum_j(I_{k,j}\ln \tau_{k,j}^{I'} \\
&+ (1-I_{k,j}')\ln(1-\tau_{k,j}^{I'}))\} = \alpha \\
\Rightarrow &\frac{1}{l_d}\textstyle\sum_k q_{k,c}\cdot\frac{1}{\pi_c} = \alpha \Rightarrow \pi_c = \frac{1}{l_d\cdot\alpha}\sum_k q_{k,c} = \frac{1}{l_d}\sum_k q_{k,c}\ (s.t.,\ \alpha=1)
\end{aligned}
\tag{30}
$$

For $\nabla_{\tau_{i,c}^A} h = 0$, we can get,

$$
\begin{aligned}
&\nabla_{\tau_{i,c}^A} (\mathcal{L}_g - \alpha\textstyle\sum_c \pi_c) = 0 \\
\Rightarrow &\nabla_{\tau_{i,c}^A} \mathcal{L}_g = 0 \\
\Rightarrow &\nabla_{\tau_{i,c}^A} \frac{1}{l_d}\textstyle\sum_{k,c} q_{k,c}\{\ln \pi_c + \sum_i(A_{i,k}\ln \tau_{i,k}^A + (1-A_{i,k})\ln(1-\tau_{i,k}^A)) + \sum_i(I_{i,k}\ln \tau_{i,k}^I + \\
&(1-I_{i,k})\ln(1-\tau_{i,k}^I)) + \sum_j(A_{k,j}\ln \tau_{k,j}^{A'} + (1-A_{k,j}')\ln(1-\tau_{k,j}^{A'})) + \sum_j(I_{k,j}\ln \tau_{k,j}^{I'} \\
&+ (1-I_{k,j}')\ln(1-\tau_{k,j}^{I'}))\} = 0 \\
\Rightarrow &\textstyle\sum_k q_{k,c} \sum_i(\frac{A_{i,c}}{\tau_{i,c}^A} - \frac{1-A_{i,c}}{1-\tau_{i,c}^A}) = 0 \\
\Rightarrow &\textstyle\sum_k q_{k,c} \sum_i \frac{A_{i,c}-\tau_{i,c}^A}{\tau_{i,c}^A\cdot(1-\tau_{i,c}^A)} = 0 \Rightarrow \tau_{i,c}^A = \frac{\sum_k A_{i,k}q_{k,c}}{\sum_k q_{k,c}}
\end{aligned}
\tag{31}
$$

For $\nabla_{\tau_{i,c}^I} h = 0$, we can get,

$$
\begin{aligned}
&\nabla_{\tau_{i,c}^I} (\mathcal{L}_g - \alpha\textstyle\sum_c \pi_c) = 0 \\
\Rightarrow &\nabla_{\tau_{i,c}^I} \mathcal{L}_g = 0 \\
\Rightarrow &\nabla_{\tau_{i,c}^I} \frac{1}{l_d}\textstyle\sum_{k,c} q_{k,c}\{\ln \pi_c + \sum_i(A_{i,k}\ln \tau_{i,k}^A + (1-A_{i,k})\ln(1-\tau_{i,k}^A)) + \sum_i(I_{i,k}\ln \tau_{i,k}^I + \\
&(1-I_{i,k})\ln(1-\tau_{i,k}^I)) + \sum_j(A_{k,j}\ln \tau_{k,j}^{A'} + (1-A_{k,j}')\ln(1-\tau_{k,j}^{A'})) + \sum_j(I_{k,j}\ln \tau_{k,j}^{I'} \\
&+ (1-I_{k,j}')\ln(1-\tau_{k,j}^{I'}))\} = 0 \\
\Rightarrow &\textstyle\sum_k q_{k,c} \sum_i(\frac{I_{i,c}}{\tau_{i,c}^I} - \frac{1-I_{i,c}}{1-\tau_{i,c}^I}) = 0 \\
\Rightarrow &\textstyle\sum_k q_{k,c} \sum_i \frac{I_{i,c}-\tau_{i,c}^I}{\tau_{i,c}^I\cdot(1-\tau_{i,c}^I)} = 0 \Rightarrow \tau_{i,c}^I = \frac{\sum_k I_{i,k}q_{k,c}}{\sum_k q_{k,c}}
\end{aligned}
\tag{32}
$$

For $\nabla_{\tau_{i,c}^{A'}} h = 0$, we can get,

$$\nabla_{\tau_{i,c}^{A'}} (\mathcal{L}_g - \alpha \sum_c \pi_c) = 0$$

$$\Rightarrow \nabla_{\tau_{i,c}^{A'}} \mathcal{L}_g = 0$$

$$\Rightarrow \nabla_{\tau_{i,c}^{A'}} \frac{1}{l_d} \sum_{k,c} q_{k,c} \{\ln \pi_c + \sum_i (A_{i,k} \ln \tau_{i,k}^A + (1 - A_{i,k}) \ln(1 - \tau_{i,k}^A)) + \sum_i (I_{i,k} \ln \tau_{i,k}^I +$$

$$(1 - I_{i,k}) \ln(1 - \tau_{i,k}^I)) + \sum_j (A_{k,j} \ln \tau_{k,j}^{A'} + (1 - A_{k,j}') \ln(1 - \tau_{k,j}^{A'})) + \sum_j (I_{k,j} \ln \tau_{k,j}^{I'}$$

$$+ (1 - I_{k,j}') \ln(1 - \tau_{k,j}^{I'}))\} = 0 \tag{33}$$

$$\Rightarrow \sum_k q_{k,c} \sum_j (\frac{A_{c,j}'}{\tau_{c,j}^{A'}} - \frac{1 - A_{c,j}'}{1 - \tau_{c,j}^{A'}}) = 0$$

$$\Rightarrow \sum_k q_{k,c} \sum_j \frac{A_{c,j}' - \tau_{i,c}^{A'}}{\tau_{i,c}^{A'} \cdot (1 - \tau_{c,j}^{A'})} = 0 \Rightarrow \tau_{c,j}^{A'} = \frac{\sum_k A_{i,k}' q_{k,c}}{\sum_k q_{k,c}}$$

For $\nabla_{\tau_{i,c}^{I'}} h = 0$, we can get,

$$\nabla_{\tau_{i,c}^{I'}} (\mathcal{L}_g - \alpha \sum_c \pi_c) = 0$$

$$\Rightarrow \nabla_{\tau_{i,c}^{I'}} \mathcal{L}_g = 0$$

$$\Rightarrow \nabla_{\tau_{i,c}^{I'}} \frac{1}{l_d} \sum_{k,c} q_{k,c} \{\ln \pi_c + \sum_i (A_{i,k} \ln \tau_{i,k}^A + (1 - A_{i,k}) \ln(1 - \tau_{i,k}^A)) + \sum_i (I_{i,k} \ln \tau_{i,k}^I +$$

$$(1 - I_{i,k}) \ln(1 - \tau_{i,k}^I)) + \sum_j (A_{k,j} \ln \tau_{k,j}^{A'} + (1 - A_{k,j}') \ln(1 - \tau_{k,j}^{A'})) + \sum_j (I_{k,j} \ln \tau_{k,j}^{I'}$$

$$+ (1 - I_{k,j}') \ln(1 - \tau_{k,j}^{I'}))\} = 0 \tag{34}$$

$$\Rightarrow \sum_k q_{k,c} \sum_j (\frac{A_{c,j}'}{\tau_{c,j}^{A'}} - \frac{1 - A_{c,j}'}{1 - \tau_{c,j}^{A'}}) = 0$$

$$\Rightarrow \sum_k q_{k,c} \sum_j \frac{I_{c,j}' - \tau_{c,j}^{I'}}{\tau_{c,j}^{I'} \cdot (1 - \tau_{c,j}^{I'})} = 0 \Rightarrow \tau_{c,j}^{I'} = \frac{\sum_k I_{i,k}' q_{k,c}}{\sum_k q_{k,c}}$$

Therefore, we get $q_{k,c}$, $\pi_c$, $\tau_{i,k}^A$, $\tau_{i,k}^I$, $\tau_{k,j}^{A'}$ and $\tau_{k,j}^{I'}$ as Eq. 35.

$$q_{k,c} = \frac{p_{k,c}}{\sum_s p_{k,c}}, \pi_c = \frac{\sum_k q_{k,c}}{l_d}, \tau^A = \frac{\sum_k A_{i,k} q_{k,c}}{\sum_k q_{k,c}}$$

$$\tau^I = \frac{\sum_k I_{i,k} q_{k,c}}{\sum_k q_{k,c}}, \tau^{A'} = \frac{\sum_k A_{k,j}' q_{k,c}}{\sum_k q_{k,c}}, \tau^{I'} = \frac{\sum_k I_{k,j}' q_{k,c}}{\sum_k q_{k,c}} \tag{35}$$

## A.4 DETAILS OF EM ALGORITHM IN INNERSIGHTNET

In this study, we propose a model based on the EM algorithm aimed at discovering potential community structures in the data. The EM algorithm is an iterative algorithm used for parameter estimation and inference of potential data structures, particularly suitable when the model contains unobservable hidden variables, as described in **appendix A.2**.

Firstly, we defined the function E Step for the expected step (E step). In step E, based on the current model parameter estimation, calculate the expected value of the latent variable $q_{k,c}$. The logarithmic probability form is used in the calculation to avoid numerical instability when dealing with extremely small values. Specifically, the model parameters include: $\{\pi_c, \tau_{i,k}^A, \tau_{i,k}^I, \tau_{k,j}^{A'}, \tau_{k,j}^{I'}\}$.

Subsequently, we implemented the function M step for maximizing step (M step). In step M, update the model parameters based on the expected values of the latent variables obtained in step E, in order to maximize the logarithmic likelihood function of the observed data.

As is well known, the EM algorithm has instability and is prone to getting stuck in local optima. In the process of executing the EM algorithm, in order to alleviate the above problems, we have adopted three main strategies: random initialization, multiple start strategy, and converting multiplication operations into logarithmic operations. The following will provide a detailed description of the implementation methods and their purpose and role of these strategies.

**(a) Random initialization:** Random initialization refers to randomly assigning model parameters before the EM algorithm starts. This is because the EM algorithm, as a gradient based optimization method, relies heavily on the initial values of the parameters to find the final solution. If the parameters are initialized properly, the algorithm is more likely to converge to the global optimal solution or a better local optimal solution. On the contrary, improper initialization may lead to slow convergence speed or suboptimal solutions for the algorithm. Through random initialization, we can explore the parameter space from multiple different starting points, increasing the chances of finding better solutions.

**(b) Multiple start strategy:** Multiple start strategy refers to repeatedly executing the algorithm multiple times, each time using different random initialization parameters. This strategy is based on the assumption that by independently starting optimization from different initial points multiple times, we can select the best local optimal solution from multiple found ones, thereby reducing the risk of the algorithm falling into poor local optimal solutions. In this study, we set the number of multiple starts to 100.

**(c) Convert multiplication to logarithmic operation:** In step E of the EM algorithm, it is necessary to calculate the product of probabilities, which are often very small. Direct multiplication can lead to numerical underflow, meaning that the computer cannot represent such small values. To avoid this situation, we adopt the method of converting multiplication operations to logarithmic operations. Specifically, by utilizing the properties of logarithmic functions, multiplication can be transformed into addition: taking the logarithm of the probability, adding it up, and finally converting it back to the original probability space through exponential transformation. This conversion not only prevents numerical problems, but also improves the numerical stability of the entire calculation process due to the more stable addition operation.

### A.5 DETAILS OF ROLES AND FUNCTIONALITIES ANALYSIS IN INNERSIGHTNET

In the communities of deep neural networks, their roles and functions are more intuitively reflected in input and output. InnerSightNet provides quantitative analyses from the input-output perspective.

**Differential outcome analysis:** In order to quantitatively analyze the impact of communities on output, we adopt a differential outcome analysis. The differential outcome analysis are statistically analyzed to determine the changes in the output of the neural network between corresponding community is not closed and closed after inputting the same data.

**Perturbation-statistical analysis:** In order to quantitatively analyze the impact of input data on the community, we adopt a perturbation-statistics analysis. By perturbing the input data and recording the response changes of the community, this method allows us to calculate the sensitivity of each community towards changes in input data. We define the sensitivity of the community as $S_c = \frac{1}{N}\sum_{i=1}^{N} f(X_i, X_i^{'})$, where $N$ is the number of samples in the test set. $f$ is a function that evaluates the difference in feature representation between the original input $X_i$ and the perturbed input $X_i^{'}$.

When starting perturbation analysis, we are not limited to the perturbation of independent pixels, but extend it to a $5{\times}5$-pixel neighbourhood. This operation takes into account the correlation between adjacent pixels in the image. We define a neighborhood perturbation function $\mathrm{Per}(X_i, x, y)$ that sets the pixels of an image at position $(x, y)$ and its neighborhood to 0, i.e., $\mathrm{Per}(X_i, x, y) = X_i^{'}$ where $X_i^{'}(u, v) = 0$ for $(u, v) \in N(x, y)$. To measure the impact of perturbations on the neuronal community, we calculated the mean squared error (MSE) of feature representations between the original and perturbed samples. The response of community $c$ to samples pairs $(X_i, X_i^{'})$ is:

$$f(X_i, X_i') = \text{MSE}_{X_i} \frac{1}{M} \sum_M^{j=1} (h_c(X_i)_j - h_c(X_i')_j)^2 \tag{36}$$

where $M$ is the number of neurons. $h_c(X_i)$ and $h_c(X_i')$ represent the feature representations of $X_i$ and $X_i'$. Then we calculate the root of $f(X_i, X_i')$. Perturbation-statistical analysis traverses the input image. Each pixel represents the overall response of the input data to community $c$ at point $(x, y)$. We obtain perturbation-statistical analysis results that are consistent with the size of the input data.

We provide a detailed introduction to perturbation-statistical analysis here. Perturbation-statistical analysis measures which regions of the input data are sensitive to a community in a deep neural network. The sensitivity of a community to input information is directly related to the flow of information and decision-making processes in deep neural networks. We obtain perturbation-statistical analysis results that are consistent with the size of the input data.

## A.6    MORE DETAILS OF SECTION 4.4

To further demonstrate the significance of InnerSightNet, we use the **Wilcoxon test** to determine the differences between the results of the methods are statistically significant. The Wilcoxon signed-rank test is a non-parametric statistical hypothesis test used either to test the location of a population based on a sample of data, or to compare the locations of two populations using two matched samples, which be applied in statical significance tests. We use MATLAB to perform Wilcoxon rank sum test.

- Zero hypothesis $H_0$: Two sets of data come from the same distribution, meaning that there is no significant difference between the two sets of data overall.
- Alternative hypothesis $H_1$: The two sets of data come from different distributions, meaning there is a significant difference between the two sets of data.

We list the results of wilcoxon rank sum test between InnerSightNet and the baselines in the following table.

Table 4: Tthe results of wilcoxon rank sum test between InnerSightNet and the baselines.

| baselines | p-value (MNIST) | Statistic (MNIST) | p-value (AFHQ) | Statistic (AFHQ) |
|---|---|---|---|---|
| Filan *et al.* | 0.000212 | -3.704051 | 0.000381 | -3.552866 |
| Hod *et al.* | 0.000157 | -3.779644 | 0.004071 | -2.872529 |
| Liu *et al.* | 0.001490 | -3.174901 | 0.001498 | -3.1749015 |

where **Statistic** represents the magnitude and direction of the difference in rank sum between two samples. The negative statistic indicates that the rank of the first set of data is generally lower than that of the second set of data. This means that the values of the first set of data are generally smaller than those of the second set. **p-value** represents the probability of observing extreme or even more extreme results under the assumption that the $H_0$ is true. Usually, when the p-value is less than the significance level (such as 0.05 or 0.01), we reject the $H_0$.

From the above table, it can be seen that the **p-values** are all less than 0.01. We can reject the $H_0$ and conclude that the two sets of data are statistically significantly different. The **Statistic** are negative values that further indicates the performances of baselines are generally lower than those of InnerSightNet.

## A.7    LIMITATIONS

Although InnerSightNet has demonstrated its unique advantages in partitioning communities based on the input-output representations of neurons, determining the best number of communities, and conducting in-depth analysis of information flow and decision-making processes in deep neural networks, the algorithm still faces two significant limitations.

Firstly, the **time consumption** of algorithms cannot be ignored. The core of InnerSightNet is based on the EM algorithm, which is an iterative optimization process. Its iterative nature itself means an increase in time cost. In order to avoid the risk of EM algorithm getting stuck in local optima, we introduce the multiple start strategy. Although this strategy improves the algorithm's global search ability, it further exacerbates the burden of computation time.

The complexity analysis of InnerSightNet reveals the root cause of its time consumption. InnerSightNet initialization stage involves setting model parameters and initializing the optimal logarithmic likelihood value, with a constant level of complexity and minimal impact on the overall performance. However, the main body of the algorithm consists of two layers of loops: the outer loop is responsible for the algorithm restart mechanism, executing $R$ times; The inner loop is responsible for iteratively optimizing the model parameters, with a maximum of T iterations executed per restart. The complexity of these two layers of loops is $O(R)$ and $O(T)$, respectively. In the inner loop, the algorithm needs to perform probability calculations and parameter updates on each of the $K$ samples and C clusters, with a complexity of $O(KC)$. Due to these operations being executed in each iteration, the overall complexity is proportional to the number of iterations T, i.e. $O(TKC)$. Taking into account the $R$ restarts of the outer loop, the overall complexity of the entire algorithm is $O(RTKC)$.

In order to reduce time consumption, we adopt multiple strategies. Firstly, we migrate the computation process to the GPU and use Cupy instead of Numpy to improve computational efficiency. Secondly, we pre calculated the average feature value, average Jacobian matrix, etc., to reduce the evaluation time for each community partition. Although these measures have to some extent alleviated the time pressure, the computation time of InnerSightNet is still relatively long. When the number of single-layer neurons is 128, the computation time for InnerSightNet in processing linear and convolutional neural networks is approximately 4 hours and 7 hours, respectively. Therefore, how to further optimize the algorithm to reduce time consumption becomes the focus of our future research.

Secondly, the issue of **concentration in community partitioning** is also worth paying attention to. When applying InnerSightNet in convolutional neural networks, we find that community partitioning is too centralized, which is in stark contrast to the situation where linear neural network analysis can partition more than 10 communities. This phenomenon raises a question: *in common sense, cat and dog images contain more information than handwritten digit, why is there actually less community division?* Our explanation is 'task-related'. Due to the fact that cat and dog classification is a binary task, the number of effective neurons for binary classification is indeed less than that for ten class tasks. In addition, our community partitioning method is based on classification results, which may lead to a bias towards classification-specific features rather than common features during the partitioning process. Therefore, developing evaluation methods suitable for non-classification networks to focus community partitioning more on detailed features, such as neurons within a community specifically responsible for recognizing cat eyes, is our future research direction and one of the ways to extend InnerSightNet to generative models.

