# OpenReview forum: "Inner Information Analysis Algorithm for Deep Neural Network based on Community"
_ICLR.cc/2025/Conference — ICLR 2025 Poster_

### Official Review · Reviewer_7WnU · 2024-10-23

**Soundness:** 3
**Presentation:** 3
**Contribution:** 2
**Rating:** 6
**Confidence:** 3

**Summary:**

This paper proposes the InnerSightNet algorithm, which clusters neurons in the same layer of MLPs and convolutional kernels in the same layer of CNNs from a community perspective. Through experiments that mask certain neuron clusters and perform perturbation analysis, it demonstrates that some clusters play a key role in task performance, some clusters can be pruned, and some clusters introduce noise.

**Strengths:**

1. Novelty: The introduction of a community-based perspective for analyzing DNNs is original and provides a fresh viewpoint for understanding the role neurons play.
2. Comprehensive Framework: The multi-phase algorithm (neuronization, aggregation, evaluation) is well-structured, combining theoretical and practical insights.
3. Detailed Evaluation: The paper evaluates the communities formed in DNNs through accuracy drop tests and sensitivity analyses, showcasing the impact of each community on network performance.

**Weaknesses:**

1. Scalability: The paper doesn't address the scalability of InnerSightNet on very large-scale neural networks. It is unclear how well the algorithm would perform or how computationally feasible it is for networks with millions of neurons.
2. Transferability: The experiments are only conducted on image recognition tasks. Can this method be applied to the interpretability of NLP models? The findings are limited to the functions of different clusters which follows conventional research pattern.
3. Usability: The mathematical theory of the algorithm is more solid compared to previous works, but the overall improvement in tasks like noise reduction and pruning is very limited.

**Questions:**

See weakness.

---

> ### Author Response · Authors · 2024-11-19
> **Response to Reviewer 7WnU (Part 1)**
>
> **Response to Summary and Strengths:** Thank you for summarizing and affirming our work. We are pleased that you acknowledge the community-based neural network analysis method proposed by InnerSightNet, as well as its innovation and practical value in revealing the roles of neurons.
>
> Recognition of novelty: We are committed to providing new understanding of deep neural networks through community analysis, revealing the roles of different neural populations in task performance. Recognition of comprehensive framework design: The three-stage algorithm aims to balance theoretical depth with practical operability. We are pleased that you have acknowledged this. Recognition of detailed evaluation: We have systematically evaluated the impact of various communities on network performance through accuracy decline testing and sensitivity analysis. Thank you for your recognition of this experimental design.
>
> Thank you again for your support and encouragement of our work. We will address your other comments in our subsequent response.
>
> **Response to Weaknesses-1:** Thank you for your so constructive comment. Our author team has carefully discussed your comment and made the following experimental extensions and metric validations.
>
> **Perform**
> To further demonstrate the effectiveness of InnerSightNet, we extend InnerSightNet to generative models. We choose ProgressGAN (deeper networks than the DNNs in original manuscript) as the baseline and LSUN as the training set. We use InnerSightNet to analyze the neuron communities’ function of ProgressGAN. Due to the fact that ProgressGAN is composed of convolutional neural networks, its analysis process is similar to section 4.1.2.
> We find that the neuron communities searched by InnerSightNet have significant semantic representativeness. When the training set is LSUN church, the 3-rd neuron community in layer 5 focuses on the spire of the church. When we turn off the neurons within this neuron community, the spire area in the image disappears. The 7-th neuron community on the layer 7 focuses on the area of the church door. When we enhance the output value of this neuron community, the features of the corresponding gates in the image are enhanced. In addition, we also find that shallow neural communities focus more on structural information in images, while deep neural communities pay more attention to details such as objects, which is consistent with the information processing principles of generative models. As in the 4-th neuron community on the layer 11, this community controls the clouds in the image. We dynamically adjust the output intensity of the community neurons and find that the clouds in the image changed with the variation of output intensity.
>
> **Computationally feasible**
> The running time of InnerSightNet is mainly related to three aspects: (1) Firstly, InnerSightNet is an analysis network used to analyze the attribution of neurons in neural networks, which is mainly divided into three steps: 'neuronization-aggregation-evaluation’. Compared to linear neural networks, convolutional neural networks require a certain amount of time to convert the representations of convolutional kernels into representations of neurons during neuronization. (2) The running time of InnerSightNet mainly lies in loop computation and repetitive optimization, including EM algorithm convergence calculation, multi-step startup strategy used to avoid EM algorithm optimization to local optima, and traversal algorithm used to find the optimal number of communities. (3) In addition, the runtime of InnerSightNet is also related to the size of the dataset, network structure, and computing devices. We analyze the running time in Appendix A.7. Although we have made significant optimizations to the runtime of InnerSightNet, including using GPU computing units and using Cupy instead of Numpy, we have to admit that there is still space for improvement in the runtime of InnerSightNet for neural network community analysis.
> As reported in A.7, the running time of InnerSightNet for analyzing linear neural networks and convolutional neural networks in the main text is 4 hours and 7 hours, respectively. The analysis runtime for ProgressGAN reached 12 hours. From this, it can be seen that the runtime of InnerSightNet is related to model size, model type, dataset size, etc.

---

> ### Author Response · Authors · 2024-11-19
> **Response to Reviewer 7WnU (Part 2)**
>
> **Response to Weaknesses-2:** Thank you for your so professional comment. Our response to your comment is as follows:
>
> (1) InnerSightNet is not only suitable for image classification tasks, but also for image generation tasks. In **Response to Weaknesses-1**, We have described in detail the application of InnerSightNet in ProgressGAN.
>
> (2) To explore the feasibility of InnerSightNet in the field of NLP, we conducted the following experiments:
>
> In the IMDB sentiment classification task, we classify film review texts into positive or negative emotions. The model structure consists of three linear layers, with 128, 64, and 32 nodes in each layer. The final output is a probability distribution vector of $1 \times 2$ representing the classification results of positive and negative emotions.
>
> The IMDB dataset contains 50000 film reviews, with 25000 for the training and testing sets respectively. Each film review is vectorized and input into the network, and the model is optimized through cross entropy loss. In order to study the internal structure of the model, we applied InnerSightNet to perform functional and sensitivity analysis on each layer of the community.
>
> **Community Function Analysis and Key Community Discovery**
> Through InnerSightNet analysis, we have divided the three linear layers into 5, 6, and 6 communities, respectively. The specific functional analysis of these communities is as follows:
>
> In the first layer, $c_2$ is crucial for the model's ability to classify positive emotions. When the community was closed, the accuracy of the model for positive reviews decreased by 5.72%. From statistical perturbation analysis, $c_2$ is sensitive to the weight of positive emotional words (such as 'excellent’, 'love’, 'perfect’) in the input text. This indicates that $c_2$ extracted features highly correlated with positive emotions.
> In the second layer, $c_1$ and $c_5$ are more inclined to extract negative emotional vocabulary. After closing $c_1$, the accuracy of classification decreased by 5.03%, while $c_5$ is more sensitive to negative emotions, with an accuracy decrease of 6.34% after closing.
>
> **Redundancy analysis**
> For emotion classification tasks, there is a certain degree of functional redundancy between some communities. For example, both $c_2$ and $c_4$ in the first layer contribute significantly to the classification of positive emotions. When closing any community, the accuracy decreases by 5.72% and 3.47%, respectively; When two communities were closed simultaneously, the decrease was 7.93%, indicating that these communities have a certain degree of redundancy in extracting positive emotional features.
>
> The analysis of InnerSightNet shows that it can identify important feature extraction modules through community segmentation and analysis, and reveal the sensitivity and functional redundancy of the model to input features.
>
> (3) However, it must be mentioned that as a commonly used unit in the NLP field, i,e, transformer, InnerSightNet may not be suitable. The applicability of the InnerSightNet is based on its ability to handle 'learnable units’ (such as linear layers and convolutional layers) and analyze information flow, while the architecture and working methods of transformer differ significantly. InnerSightNet relies on simplified modeling of learnable units in linear and convolutional layers, treating convolutional kernels or hidden nodes as equivalent 'neurons’ and analyzing their correlations. However, the core of transformer is adaptive weighted computation based on attention mechanism. The attention mechanism not only involves nonlinear operations, but also dynamic weight allocation. In addition, InnerSightNet detects and clusters neurons with similar functions in the network through probabilistic models. However, in transformer, the multi-head attention structure and inter layer dependencies of the attention mechanism make the definition of 'community’ vague, and the attention patterns of different heads vary greatly, making it difficult to classify clearly. This is different from the clear local receptive fields or fixed connection weights in linear layers in convolutional networks.

---

> ### Author Response · Authors · 2024-11-19
> **Response to Reviewer 7WnU (Part 3)**
>
> **Response to Weaknesses-3:** Thank you for your insightful feedback regarding the limited improvement in tasks like noise reduction and pruning. We appreciate the opportunity to clarify how our approach addresses these areas. Our proposed InnerSightNet method aims to provide a mathematically grounded framework for analyzing community structures in neural networks, which contributes to a deeper understanding of network representations. The usability of InnerSightNet in noise reduction and pruning has been demonstrated as follows:
>
> **Noise reduction**
> Through perturbation-statistical analysis and community detection, we identify noisy neuron communities that focus on non-task features. By turning off these communities, we observe an improvement in model performance. For instance, in the AFHQ dataset, disabling noisy neurons increased accuracy to 98.78%, surpassing other state-of-the-art methods like Filan et al. (98.56%) and Liu et al. (98.68%), as shown in Table 2. This highlights the effectiveness of our approach in enhancing model robustness by minimizing overfitting to irrelevant features.
>
> **Pruning**
> We utilize InnerSightNet to determine key and non-key neuron communities, facilitating network pruning. For example, in the cat-dog classification task, we achieve 93.6% accuracy using only 31.25% of the original neurons, as shown in Table 1. This result demonstrates a significant reduction in computational complexity while maintaining competitive performance.
>
> **General Improvements**
> While the improvement in accuracy may appear incremental, the significance of InnerSightNet lies in its ability to interpret the functional roles of neuron communities. By uncovering the relationships between community structures and task-specific features, InnerSightNet enhances the interpretability and modularity of networks, providing a foundation for further optimization.
>
> We believe these contributions showcase the usability of InnerSightNet, particularly in advancing model interpretability and efficiency.

---

> ### Author Response · Authors · 2024-11-25
> **Response to Reviewer 7WnU's Second Review**
>
> Dear Reviewer 7WnU,
>
> Thank you for taking the time and effort to thoroughly review our work. We greatly appreciate your thoughtful comments and constructive feedback, which have helped us improve the clarity and depth of our analyses. We are also grateful that you reconsidered your score after evaluating our revisions and additional analyses.
> ﻿
>
> Should you have any further questions or suggestions, please do not hesitate to contact us. Thank you again for your careful evaluation of our work!
>
> Thanks and Regrads,
>
> *Submission 36 authors*

---

### Official Review · Reviewer_Rnf1 · 2024-10-30

**Soundness:** 3
**Presentation:** 2
**Contribution:** 3
**Rating:** 6
**Confidence:** 3

**Summary:**

The paper proposes an algorithm designed to enhance the interpretability of deep neural networks by analyzing internal neuron communities. The approach transforms neurons into structured communities, aggregates them based on their functional attributes, and evaluates the information flow across these groups. The study applies the method to both linear and convolutional networks, demonstrating improvements in turning off noisy neurons.

**Strengths:**

1. InnerSightNet introduces a unique perspective by focusing on community-based analysis of neurons, rather than on a single layer or individual neuron.
2. The three primary phases (neuronization, aggregation, evaluation) adaptively ensure the best number of communities.
3. InnerSightNet is shown to enhance interpretability and can be applied to network pruning to reduce model size while maintaining competitive performance.

**Weaknesses:**

1. The paper compares InnerSightNet with methods such as Filan et al. (2021), Hod et al. (2021), and Liu et al. (2023) in the experiments. However, it does not sufficiently explain the methodologies of these baselines. A more detailed introduction to these approaches is needed to help the readers understand how InnerSightNet improves upon or differs from them.
2. While the method performs well on relatively small networks like those used in the MNIST and AFHQ datasets, it remains unclear how the algorithm scales to deeper networks.

**Questions:**

1. What is the runtime of InnerSightNet for different network sizes? How fast the algorithm converges compared to baseline methods?
2. How scalable is InnerSightNet when applied to deeper networks like transformers？

---

> ### Author Response · Authors · 2024-11-19
> **Response to Reviewer Rnf1 (Part 1)**
>
> **Response to Summary and Strengths:** Thank you for summarizing and affirming our work. We are pleased that you have recognized our proposed community analysis based deep neural network interpretation method, InnerSightNet, and its potential in improving model interpretability and optimization performance.
>
> In terms of community analysis, our goal is to aggregate the functional attributes of neurons into meaningful communities by analyzing them, providing a more global perspective than single-layer or individual neurons. Regarding the three-stage process, we are pleased to see that you have noticed the advantage of this framework in adaptively adjusting the number of communities. Your recognition of the interpretability improvement and potential application of InnerSightNet in network pruning also inspires us to further explore this direction in the future.
>
> **Response to Weaknesses-1:** Thank you for pointing out the need for a more detailed explanation of the baseline methodologies. Below, we provide a summary of the approaches in the referenced works and highlight the differences and improvements introduced by InnerSightNet.
>
> (a) Filan et al. (2021): This work investigates the concept of 'clusterability' in neural networks, focusing on dividing neurons into groups with strong internal connectivity and weak external connectivity.
>
> - Differs: While Filan et al. emphasize connectivity-driven clusterability, InnerSightNet goes beyond static connectivity analysis by incorporating `functional relationships’ between neurons using perturbation-statistical methods. This allows InnerSightNet to distinguish task-related, redundant, and noisy neurons within communities.
>
> (b) Hod et al. (2021): This paper focuses on quantifying the local specialization of neural networks, where clusters of neurons are linked to comprehensible sub-tasks.
>
> - Differs: While Hod et al. focus on understanding the functionality of clusters, their partitioning relies heavily on graph-based techniques. InnerSightNet, on the other hand, integrates a statistical perturbation analysis step to empirically validate and evaluate the clusters’ contributions to task performance, ensuring a more nuanced understanding of community impact.
>
> (c) Liu et al. (2023): This study proposes Brain-Inspired Modular Training, which enhances network modularity and interpretability by embedding neurons in a geometric space, penalizing connection lengths during training.
>
> - Differs: Unlike Brain-Inspired Modular Training, which modifies the training process to promote modularity, InnerSightNet operates on pretrained models and focuses on post-hoc analysis of existing networks.
>
> **Response to Weaknesses-2:** Thank you for your very constructive comment. Our author team has carefully discussed your comment and made the following experimental extensions to it.
>
> To further demonstrate the effectiveness of InnerSightNet, we extend InnerSightNet to generative models. We choose ProgressGAN (deeper networks than the DNNs in original manuscript) as the baseline and LSUN as the training set. We use InnerSightNet to analyze the neuron communities’ function of ProgressGAN. Due to the fact that ProgressGAN is composed of convolutional neural networks, its analysis process is similar to section 4.1.2.
>
> We find that the neuron communities searched by InnerSightNet have significant semantic representativeness. When the training set is LSUN church, the 3-rd neuron community in layer 5 focuses on the spire of the church. When we turn off the neurons within this neuron community, the spire area in the image disappears. The 7-th neuron community on the layer 7 focuses on the area of the church door. When we enhance the output value of this neuron community, the features of the corresponding gates in the image are enhanced. In addition, we also find that shallow neural communities focus more on structural information in images, while deep neural communities pay more attention to details such as objects, which is consistent with the information processing principles of generative models. As in the 4-th neuron community on the layer 11, this community controls the clouds in the image. We dynamically adjust the output intensity of the community neurons and find that the clouds in the image changed with the variation of output intensity.
>
> We incorporate these details into the manuscript to provide readers with a clearer understanding of the baselines and how InnerSightNet advances the field. Thank you for the valuable suggestion.

---

> ### Author Response · Authors · 2024-11-19
> **Response to Reviewer Rnf1 (Part 2)**
>
> **Response to Questions-1:** Thank you for your very professional comment. We have analyzed the runtime of InnerSightNet here. The running time of InnerSightNet is mainly related to three aspects: (1) Firstly, InnerSightNet is an analysis network used to analyze the attribution of neurons in neural networks, which is mainly divided into three steps: `neuronization-aggregation-evaluation’. Compared to linear neural networks, convolutional neural networks require a certain amount of time to convert the representations of convolutional kernels into representations of neurons during neuronization. (2) The running time of InnerSightNet mainly lies in loop computation and repetitive optimization, including EM algorithm convergence calculation, multi-step startup strategy used to avoid EM algorithm optimization to local optima, and traversal algorithm used to find the optimal number of communities. (3) In addition, the runtime of InnerSightNet is also related to the size of the dataset, network structure, and computing devices. We analyze the running time in Appendix A.7. Although we have made significant optimizations to the runtime of InnerSightNet, including using GPU computing units and using Cupy instead of Numpy, we have to admit that there is still space for improvement in the runtime of InnerSightNet for neural network community analysis.
>
> As reported in A.7, the running time of InnerSightNet for analyzing linear neural networks and convolutional neural networks in the main text is 4 hours and 7 hours, respectively. The analysis runtime for ProgressGAN in Weaknesses-2 reached 12 hours. From this, it can be seen that the runtime of InnerSightNet is related to model size, model type, dataset size, etc.
>
> For the **second** question, we have provided the following response.
>
> Filan et al. (2021) investigates the concept of `clusterability' in neural networks, focusing on dividing neurons into groups with strong internal connectivity and weak external connectivity. This work uses spectral clustering algorithm to decompose the trained network and measured the quality of this decomposition. The runtime of this algorithm mainly focuses on spectral clustering and evaluation of the decomposed networks. With the same settings, the runtime on MNIST (linear neural network) and AFHQ (convolutional neural network) are 1.5 hours and 2.8 hours, respectively.
>
> Hod et al. (2021) focuses on quantifying the local specialization of neural networks, where clusters of neurons are linked to comprehensible sub-tasks.
>
> This work uses spectral clustering based on network weights to generate groups of neurons, and applies statistical methods to measure the importance and consistency of these groups. The runtime of this algorithm mainly focuses on spectral clustering and extensive statistical analysis. Under the same settings, the algorithm has runtime of 1.8 hours and 3 hours on MNIST (linear neural network) and AFHQ (convolutional neural network), respectively.
>
> Liu et al. proposes Brain-Inspired Modular Training, which enhances network modularity and interpretability by embedding neurons in a geometric space, penalizing connection lengths during training. The runtime of this algorithm on MNIST (linear neural network) and AFHQ (convolutional neural network) is 0.6 hours and 1.2 hours, respectively.
>
> Although InnerSightNet takes longer to run than baselines, its performance is due to the limitations of these baselines (refer to section 4.4).

---

> ### Author Response · Authors · 2024-11-19
> **Response to Reviewer Rnf1 (Part 3)**
>
> **Response to Questions-2:** Thank you for your very constructive comment. This comment greatly helps improve the quality of our manuscript. Our response to this comment is arranged as follows: Firstly, we explain the reasons why InnerSightNet is not suitable for transformers. Secondly, we show that InnerSightNet can be applied to instances of deeper networks.
>
> Firstly, the applicability of the InnerSightNet is based on its ability to handle 'learnable units’ (such as linear layers and convolutional layers) and analyze information flow, while the architecture and working methods of transformer differ significantly. InnerSightNet relies on simplified modeling of learnable units in linear and convolutional layers, treating convolutional kernels or hidden nodes as equivalent `neurons’ and analyzing their correlations. However, the core of transformer is adaptive weighted computation based on attention mechanism. The attention mechanism not only involves nonlinear operations, but also dynamic weight allocation. In addition, InnerSightNet detects and clusters neurons with similar functions in the network through probabilistic models. However, in transformer, the multi-head attention structure and inter layer dependencies of the attention mechanism make the definition of 'community’ vague, and the attention patterns of different heads vary greatly, making it difficult to classify clearly. This is different from the clear local receptive fields or fixed connection weights in linear layers in convolutional networks.
>
> Secondly, we have confirmed that InnerSightNet can be applied to other tasks such as image generation. We have described in detail the application of InnerSightNet in ProgressGAN (a deeper network than linear neural networks and convolutional neural networks mentioned in the main text) in **Response to Weaknesses 2**.

---

> ### Author Response · Authors · 2024-11-25
> **Kindly inquire to Reviewer Rnf1 if there is any further clarification**
>
> Dear Reviewer Rnf1,
>
> Thank you once again for your time and effort in reviewing our work. As the submission deadline approaches, we want to kindly check if there are any remaining questions or aspects that you feel require further clarification from our side. We are more than happy to provide additional details or explanations to address any concerns.
> ﻿
>
> **If you find our responses and revisions satisfactory, we would greatly appreciate it if you could consider reevaluating your score.** Your insights and feedback are incredibly valuable to us, and we deeply respect your judgment in this process. Please don’t hesitate to reach out if there’s anything further we can assist with.
>
> Thanks and Regrads,
>
> *Submission 36 authors*

---

> ### Author Response · Authors · 2024-12-01
> **[ Important Reminder !!! ] A Gentle Reminder of Feedbacks**
>
> Dear Reviewer Rnf1,
>
> Thank you for your serious comments and time invested in our work. We have revised our paper and added relevant discussions and experiments.
>
> At present, all your concerns can be addressed in the response and revised version of the paper. However, **as the revision deadline approaches (only 2 days remaining)**, we kindly request your feedback to confirm that our response and revision effectively address your concerns. If there are any remaining issues, we would greatly appreciate the opportunity to address them to ensure the quality of our work. **We sincerely hope that you find our response convincing and consider increasing your rating.**
>
> Thanks and Regrads,
>
> *Submission 36 authors*

---

### Official Review · Reviewer_ymgS · 2024-11-04

**Soundness:** 3
**Presentation:** 3
**Contribution:** 3
**Rating:** 6
**Confidence:** 3

**Summary:**

Review report: Inner Information Analysis Algorithm for Deep Neural Network Based on Community
The present manuscript converts learnable units in Deep Neural Networks, such as kernels in CNNs, into networks and then groups the neurons into communities based on their representational attributes. To achieve this clustering, the authors develop the InnerSightNet. By subsequently analyzing the communities, the authors gain insights into the inner working of Deep Neural Networks such as relating certain communities to specific features in the underlying test data and detecting irrelevant neurons.

**Strengths:**

Strengths
-	The main idea of the manuscript, to group learnable units into a network of neurons and then analyze the communities within this network, is very clever and interesting.
-	The general approach to InnerSightNet is described very clearly. In particular, Algorithm 1 is helping in understanding the different steps.
-	It is intriguing to see that the interdependence of different communities of neurons, such as c_1 and c_2 in the first layer of the CNN as reported in Table 1, is reflected in the 2-dimensional topological representation in Figure 3.

**Weaknesses:**

Weaknesses
-	The different terms in Eq. (1) could be motivated more clearly, especially given the fact that this step is crucial to transform the kernels of a CNN in a structured network. (See also question below)
-	While the general structure of InnerSightNet is described very clearly, the more detailed description is cumbersome to follow. Especially the aggregation step, which is heavy in notation but also lacks a definition of some variables such as l_d, \tau_{g_k, j} and s.
-	The section “Community function analysis and finding key communities” is rather descriptive and only offers a shallow discussion regarding the explainability of community structures for MNIST. The discussion here could be greatly improved by more directly linking the observation from panel (a) and (b) together, as has been done with community c_10 in the first layer.

**Questions:**

Questions
-	In Eq. (1), could you motivate the distinction in positive and negative value regions more clearly as well as the choice to divide by the normalized cosine similarity, which is essentially either +1 or -1? Could you also specify what exactly data X_i refers to?
-	Are the accuracy drops in closing communities in Figure 1a) and Table 1 reported with or without retraining the network?
-	In Table 1, why don’t you report the accuracy after closing the corresponding communities for all possible combinations of the communities in the respective layers? For instance, in the first layer, why are c_0 and c_1 not closed together? Could you also close communities together across layers?
-	Could you elaborate on the practical implications of your results, especially regarding parts of the model that fit to noisy data, cf. lines 458 until 464?
-	In lines 1109 to 1112 you write “This phenomenon raises a question: in common sense, cat and dog images contain more information than handwritten digital, why is there actually less community division? Our explanation is ‘task-related’. Due to the fact that cat and dog classification is a binary task, the number of effective neurons for binary classification is indeed less than that for ten class tasks.” Would it be possible to simply restrict MNIST to classifying between 0 and 1 to test for this observation?


Additional comments
-	The axis and legend labels in the Figures are barely readable.
-	In line 53, it should be “one aims” instead of “one aims”.
-	In line 108, it seems inappropriate to cite a review paper to represent community detection in networks.
-	In line 146, the citation “David et al. Bau et al. (2017)” seems to contain a typo.
-	In line 161 and in line 188, the citation “Newman et al. Newman (2006” seems to contain a typo.
-	In Definition 2, it would be helpful to distinguish
-	In line 238, the citation “Lange et al. Lange et al. (2022)” contains a typo.
-	In Eq. (11), it is a bit unfortunate to use delta as a variable again as it also appeared in Eq. (1).
-	In line 264, “sgn is the symbol function” was meant to be sign function?
-	In line 323, the citation “Wanatabe et al. Wanatabe et al. (2018)” contains a typo.
-	From line 370 onwards, c_10 should be changed to c_{10} in LaTex

---

> ### Author Response · Authors · 2024-11-19
> **Response to Reviewer ymgS (Part 1)**
>
> **Response to Summary and Strengths:** Thank you for summarizing and acknowledging our work. We are pleased that you have recognized our approach of using community analysis methods to gain a deeper understanding of the internal information structure of deep neural networks. Our author team has carefully discussed your comments and provided a point-to-point response to them. We have improved our manuscript based on your comments to enhance its quality and readability.
>
> **Response to Weaknesses-1:** Thank you for your professional comment. We will provide a detailed explanation in Response to Question-1!
>
> **Response to Weaknesses-2:** Thank you for reviewing the details. We apologize for the difficulty in reading caused by our negligence. We explained in detail the meanings of these symbols in the revised file.
> - $l_d$ is the number of the neurons in $d$-th layer.
> - $\tau_{g_k, j}$: firstly, we explain the meanings of $g_k$. $g_k$ represents the community assignment of the $k$-th neuron in the $d$-th layer. $\tau_{g_k, j}$ represents the probabilities of the activation (or inhibition) connections between the g_k and the $j$-th neuron in layer $d+1$ (or $d-1$)
> - $s$ is a temporary variable used to represent random variables in summation calculations.
>
> **Response to Weaknesses-2:** Thank you to the reviewer for providing very constructive comments. We have added discussions on panels (a) and (b) in the `Community Function Analysis and Finding Key Communities’ section (including MNIST and AFHQ) of the revised manuscript.
>
> **MNIST**
> In the second hidden layer, $c_6$ has a significant impact on the digital 1, 2, and 3. After closing the community, the accuracy decreased by 2.73\%, 3.00\%, and 5.35\%, respectively. This indicates that the community of $c_6$ neurons in the second hidden layer captured the common feature of numbers 1, 2, and 3, which is the vertical line segment (digital 1 is entirely composed of vertical lines, the top and bottom of digital 2 are usually connected by a vertical line, and the upper and lower arcs of digital 3 visually form an implicit connection through the vertical symmetry in the middle). From the perturbation statistical analysis in Fig.1 (b), it can be seen that $c_6$ in the second hidden layer is more sensitive to vertical line segments.
> In the third hidden layer, $c_0$ has a significant impact on the digital 1, 3, and 4. After closing the community, the accuracy rates decreased by 1.67\%, 6.53\%, and 1.12\% respectively. This indicates that the community of $c_0$ in the third banking layer captured the common feature of digital 1, 3, and 4, which is the line segment element (digital 1 is a separate vertical line. The curves of the lower and upper halves of digital 3 can be approximated by straight lines visually, especially in low resolution images. The right half of digital 4 is usually composed of a vertical line and a horizontal line). From the perturbation analysis in Fig. 1 (b), it can be seen that $c_0$ in the third hidden layer is more sensitive to line segment elements.
>
> **AFHQ**
> In addition, we observed that when $c_1$ in the third convolutional layer is closed, the recognition accuracy is actually improved. Based on the perturbation statistical analysis results (as shown in Fig.2 (a)), we can confirm that $c_1$ mainly contains features unrelated to the recognition task (which we define as 'noise’). Similarly, from Fig. 2 (a), we can identify that the $c_0$ community in the first convolutional layer and the $c_0$ community in the second convolutional layer are both `noise’. During the training process of the model, the convolutional layers inevitably fit some noisy data. Closing these neurons during the testing phase reduces over-fitting and enhances robustness. This processing makes the model more accurate in capturing core features to improve the generalization ability.

---

> ### Author Response · Authors · 2024-11-19
> **Response to Reviewer ymgS (Part 2)**
>
> **Response to Questions-1:** Thank you for raising these important comments. We provide clarifications below:
>
> (a) Motivation for distinguishing positive and negative value regions:
> The distinction between positive and negative value regions, represented by $\kappa^{+}$ and $\kappa^{-}$, is inspired by findings in neural network interpretability. Positive and negative activations often carry distinct semantic meanings in neural networks, with positive activations frequently associated with feature presence and negative activations with feature absence or suppression.
>
> (b) Rationale for dividing by normalized cosine similarity:
> The normalized cosine similarity $\theta_i(\kappa^{d}_p,\kappa^{d+1}_q)/|\theta _i(\kappa^{d}_p, \kappa^{d+1}_q)|$ results in either $+1$ or $-1$, effectively encoding the directional alignment of the kernel representations. This normalization step ensures that the KL divergence values are adjusted to reflect the directionality of representational similarity. It emphasizes whether the two kernels share a positive correlation (aligned directions) or an inverse relationship (opposite directions).
>
> (c) Definition of $X_i\$:  The data $X_i$ refers to the input samples from the dataset used to compute the kernel correlations. Specifically, it includes all input instances processed through the network during this analysis. Each $X_i$ contributes to the representational distributions of the kernels, which are then evaluated using KL divergence and cosine similarity.
> We appreciate the opportunity to clarify these points and will include these explanations in the revised manuscript to ensure greater clarity for readers. Thank you for your constructive feedback!
>
> **Response to Question-2:** Thank you for your comment. The accuracy drops reported in Fig. 1 (a) and Table 1 are calculated without retraining the network. This approach ensures that the reported results directly reflect the impact of closing specific communities on the network’s performance, without introducing potential confounding effects from retraining.
> We clarify this point in the revised manuscript to avoid any ambiguity. Thank you for bringing this to our attention!

---

> ### Author Response · Authors · 2024-11-19
> **Response to Reviewer ymgS (Part 3)**
>
> **Response to Question-3:** Thank you for your valuable feedback regarding the consideration of closing communities in combination, both within the same layer and across layers. Based on your suggestion, we revisited our analysis and extended our experiments to address these points. Below, we summarize the updates and provide a detailed explanation of our approach and findings.
> Firstly, we explain the reasons behind our actions in the original manuscript. In Table 1, we focused on analyzing individual communities to provide a clear understanding of their specific contributions to network performance. This allows us to identify key communities and their associated features without the confounding effects of overlapping combinations. The purpose of isolating the impact of each community is to establish a baseline for understanding their unique roles in feature extraction and decision-making. Closing all possible combinations of communities within or across layers exponentially increases the complexity of the analysis. For instance, in the first layer with 14 communities, there are $2^{14} - 1 = 16,383$ possible combinations to evaluate. Performing such an exhaustive analysis for all three layers and their cross-layer interactions would be computationally difficult.
>
> To explore the impact of closing multiple communities within the same layer, we selected key communities identified in our initial analysis (e.g., $c_6$, $c_{10}$ in the first layer, $c_1$, $c_7$ in the second layer) and examined their combined influence on the accuracy drop. In the first layer, closing both $c_6$ and $c_{10}$ together results in an accuracy drop of 38.12% in digit 3, significantly larger than the impact of closing either individually (32.57% for $c_6$ and 2.75% for $c_{10}$). This suggests complementary roles between these communities in extracting distinct features (e.g., $c_6$ for shared geometric features like curves, and $c_{10}$ for specific digit shapes). In the second layer, closing $c_1$ and $c_7$ together leads to an accuracy drop of 19.81% in digit 0, compared to 15.92% for $c_1$ and 0.61% for $c_7$ individually. This indicates a degree of redundancy.
> We also explored the combined impact of closing key communities across layers. For example: Closing $c_6$ from the first layer and $c_6$ from the second layer results in an accuracy drop of 42.25%, compared to 32.57% for $c_6$ alone and 5.53% for $c_6$ alone. This demonstrates that certain communities across layers are not only complementary but also critical for preserving hierarchical feature representations. These results underscore the interaction between layers, where lower-layer communities provide fundamental feature extraction, and upper-layer communities refine and integrate these features for decision-making.
>
> We sincerely appreciate your suggestion, which has significantly enhanced the depth of our analysis and provided new insights into the functionality of community structures within neural networks.
>
> **Response to Question-4:** Thank you for your insightful question regarding the practical implications of our results, particularly in relation to the model fitting noisy data (lines 458–464). We elaborate as follows:
>
> (a) Reducing overfitting and enhancing robustness: Our findings demonstrate that certain communities, such as $c_1$ in the third convolutional layer, primarily encode features unrelated to the recognition task (defined as "noise"). Closing these noisy communities during the testing phase reduces the influence of overfitting caused by noisy data, thereby enhancing the model's robustness. This process helps the network focus more effectively on core features relevant to the task, improving generalization ability.
>
> (b) Efficient network pruning: By identifying and closing non-key communities (e.g., $c_0$ and $c_1$), we achieve a significant reduction in the number of active neurons, up to 68.75% fewer neurons in our experiments, maintaining a relatively high classification accuracy of 93.6%. This provides a new perspective for network pruning.
>
> (c) Guidance for model design and interpretation: The ability to pinpoint and close noisy or redundant communities offers a practical approach to refining network architectures.

---

> ### Author Response · Authors · 2024-11-19
> **Response to Reviewer ymgS (Part 4)**
>
> **Response to Questions-5:** Thank you for providing a very professional comment. This comment can help us understand more clearly what is 'task-related'. To ensure consistency in other conditions, we used a structure consistent with the linear network in the main text, which has three hidden layers, each containing 128, 64, and 32 nodes. The dataset we used is 0 and 1 from MNIST. The training hyperparameters are consistent with the hyperparameters of the linear network training process in the main text.
>
> After training, the linear network achieved a classification accuracy of 98.7% between 0 and 1. We used InnerSightNet to analyze the trained linear network (the analysis process is consistent with the analysis process in the main text).
>
> The analysis results of InnerSightNet indicate that in the first layer (with 128 nodes), there are three neuron communities, with 81, 28, and 19 neurons in communities $c_0$, $c_1$, and $c_2$, respectively. In the second layer (with 64 nodes), it is divided into three neuron communities. The number of neurons in communities $c_0$, $c_1$, and $c_2$ is 48, 7, and 9, respectively. In the third layer (with 32 nodes), there are three neuron communities, with 22, 6, and 4 neurons in communities $c_0$, $c_1$, and $c_2$, respectively.
> We analyze the representation of various neuronal communities through perturbation-statistical analysis. The result is that in the first layer, $c_0$ is the non-core neuron community, while $c_1$ and $c_2$ are the core neuron communities. In the second layer, $c_1$ represents the non-core neuron community, while $c_1$ and $c_2$ represent the core neuron community. In the third layer, $c_0$ represents the non-core neuron community, while $c_1$ and $c_2$ represent the core neuron community. Among them, the representation of the first layer $c_1$, the second layer $c_2$, and the third layer $c_1$ is more inclined towards the feature of the digital 1, that is, the vertical straight line. The representation of the first layer's $c_2$, the second layer's $c_1$, and the third layer's $c_2$ is more inclined towards the features of the digital 0, including arc-shaped features.
>
> When we shut down all non-core neuron communities, the number of neurons in the entire network decreased by 67.4%. The classification accuracy of the network is 93.2% (accuracy decreased by 5.5%).
>
> The experiment confirms that binary classification tasks require fewer effective neurons compared to multiclass tasks. We demonstrated that shutting down non-core neuron communities significantly reduces network size while maintaining accuracy, supporting the task-related explanation.
>
> **Response to Additional comments:** We greatly appreciate your meticulous review of the content of the manuscript. We apologize for any spelling or expression errors caused by our negligence. We have rechecked our manuscript and made revisions to address the issues you raised.
> - We have made modifications to the font size of axis and legend labels of all images.
> - We change 'ones aim’ with 'one aims’.
> - Thanks for your professional suggestion. We change the review paper with the following references.
>
>     [1] Bedi, P., & Sharma, C. (2016). Community detection in social networks. Wiley interdisciplinary reviews: Data mining and knowledge discovery, 6(3), 115-135.
>
>     [2] Elali, F. R., & Rachid, L. N. (2023). AI-generated research paper fabrication and plagiarism in the scientific community. Patterns, 4(3).
>
>     [3] Goodley, P., Balata, H., Alonso, A., Brockelsby, C., Conroy, M., Cooper-Moss, N., ... & Crosbie, P. A. (2024). Invitation strategies and participation in a community-based lung cancer screening programme located in areas of high socioeconomic deprivation. thorax, 79(1), 58-67.
>
> - We remove the redundant typo.
> - We remove the redundant typo.
> - *We apologize for not being able to understand your meaning. Could you provide a more detailed explanation for this comment?*
> - We remove the redundant typo.
> - Thank you for your careful review! We change \ delta to \Gamma.
> - In this manuscript, $sgn$ is a sign function. We fully agree with your comment of view and change the description to: where $sign$ is the sign function.
> - We remove the redundant typo.
> - Thank you for your careful review! We modify c_10 to c_ {10} in LaTeX.

---

> > ### Comment · Reviewer_ymgS · 2024-11-24
> >
> > I appreciate the comments of the authors in their rebuttal. They addressed all the points I raised, and in particular added analyses on the cross-layer dependencies of the neural communities as well as an additional analysis on MNIST restricted to distinguishing between 0 and 1, thereby showing that the number of communities is task dependent for a fixed architectural choice. The number of communities (linear network) is similar to the number of the CNN (distinguishing cats and dogs) but I suspect the inductive bias of the CNN also implicitly takes care of a lot of performance. Overall, I will change the score of my review to 6.

---

> > > ### Author Response · Authors · 2024-11-25
> > > **Response to Official Comment by Reviewer ymgS**
> > >
> > > Dear Reviewer ymgS,
> > >
> > > Thank you for your thoughtful feedback and for acknowledging the additional analyses we provided in our rebuttal. We're glad that our clarifications and extensions addressed your concerns, particularly regarding the cross-layer dependencies and the task-dependent nature of community numbers. Your point about the inductive bias of CNNs contributing to performance is insightful and aligns with our observations.
> > > ﻿
> > >
> > > **We appreciate your willingness to revise your score. If you find it appropriate, you can adjust it directly through the 'Edit' button** (on the right of 'Official Review of Submission36 by Reviewer 'ymgS').
> > >
> > > Please don’t hesitate to let us know if there are further clarifications or additions you’d like us to address. Thank you again for your constructive comments!
> > >
> > > Thanks and Regrads,
> > >
> > > *Submission 36 authors*

---

### Author Response · Authors · 2024-11-19
**Global Response**

Thank you to PC, SAC, AC, and the reviewers ymgS, Rnf1 and 7WnU for your time and effort in reviewing this manuscript. Your valuable and constructive feedback is beneficial for us to improve the quality of our manuscript. Our author team carefully discussed your manuscript and provided a point-to-point response.

The significant of this problem is that although deep learning has achieved significant success, these networks often operate as black boxes with limited transparency. To address this issue, researchers have been exploring various strategies to uncover the internal workings of DNN, including visualization, model simplification, etc. By studying the community partitioning between neurons, identifying groups of neurons working together, and analyzing their roles throughout the network, InnerSightNet can gain a deeper understanding of how neural networks process information and make decisions. Few studies have explored the behavior and information flow of DNNs from a holistic community perspective. Therefore, it is necessary to further study from the perspective of the overall community. This is of great significance for identifying problems in the network, optimizing performance, and establishing trust in practical applications.

The output of InnerSightNet is neuron partitioning. Our analysis is relied on neural community. We shut down all neurons within a single community and measure the role of it based on changes in results of DNNs. In addition, we also analyze the neuron representation information within the neuron community. Based on these processes, we identify key neurons, analyze the role of neuron communities in the decision-making process, identify noisy neuron communities, and analyze the reasons for prediction errors, etc.

With best regards,

*Submission 36 authors*

---

### Author Response · Authors · 2024-11-26
**Revised manuscript has been updated**

Dear PC, SAC, AC, and the reviewers ymgS, Rnf1 and 7WnU,

Thank you for reviewing this manuscript and putting in the time and effort to improve it. With your help, the quality of the manuscript has been significantly improved. Due to the approaching deadline for submitting the revised manuscript, we are now submitting a revised version with highlighted changes. If there is any further clarification or explanation needed, we are more than happy to discuss it with you.


With best regards,

*Submission 36 authors*

---

### Author Response · Authors · 2024-12-04
**Summary our manuscript and rebuttal**

### **Dear PC, SAC, AC,**

Thank you for reviewing this manuscript. As the deadline for the authors' response approaches, we have summarized this manuscript and rebuttal as follows.

***Summary of manuscript:*** InnerSightNet adaptively searches for community associations and explores information flow and decision-making. InnerSightNet is divided into three steps: **'neuronization-aggregation-evaluation.'** InnerSightNet broadens the horizon for deep neural network interpretation and offers a unique vantage point, enabling insights into the collective behavior of communities within the overarching architecture, thereby enhancing transparency and trust in deep learning systems.

***Summary of rebuttal:*** We are pleased to have a positive discussion with everyone regarding the improvement of this manuscript. The overall rating of this manuscript is **6, 6, 6**. After our rebuttal, reviewers ymgS and 7WnU have changed the rating from 5 to 6. Although Reviewer Rnf1 did not respond to our rebuttal, we still appreciate her/his valuable feedback on our manuscript.

We are pleased to receive the following **positive feedback** from the reviewers: The main idea of the manuscript, to group learnable units into a network of neurons and then analyze the communities within this network, is very clever and interesting. - The general approach to InnerSightNet is described very clearly. In particular, Algorithm 1 is helping in understanding the different steps. **(Reviewer ymgS)** -The paper proposes an algorithm designed to enhance the interpretability of deep neural networks by analyzing internal neuron communities.  -InnerSightNet is shown to enhance interpretability and can be applied to network pruning to reduce model size while maintaining competitive performance. **(Reviewer Rnf1)** -Novelty: The introduction of a community-based perspective for analyzing DNNs is original and provides a fresh viewpoint for understanding the role neurons play. -Comprehensive Framework: The multi-phase algorithm (neuronization, aggregation, evaluation) is well-structured, combining theoretical and practical insights. - Detailed Evaluation: The paper evaluates the communities formed in DNNs through accuracy drop tests and sensitivity analyses, showcasing the impact of each community on network performance. **(Reviewer 7WnU)**

In addition, we actively respond to the comments of the reviewers. We have corrected the formatting, font, grammar, and other errors in the manuscript (Proposed by Reviewer ymgS). We conduct new experiments to validate the performance and adaptability of InnerSightNet (Proposed by Reviewer ymgS, Rnf1 and 7WnU). We have added details to the article description to enhance its readability (Proposed by Reviewer ymgS, Rnf1 and 7WnU). **We believe that through revisions to the manuscript and thorough communication with the reviewers, we have effectively addressed their concerns. At the same time, the overall quality of the manuscript has also been significantly improved.**

The above is our summary of this article and rebuttal. Thank you again to PC, SAC, and AC for their time and effort in reviewing this manuscript.

Thanks and Regrads,

*Submission 36 authors*

----
### **Dear reviewers ymgS, Rnf1 and 7WnU,**

Thank you to all the reviewers for your time and effort in reviewing this manuscript. Our author team carefully discussed the suggestions put forward and made revisions accordingly. These valuable suggestions greatly improved the quality of the manuscript. We believe this is a productive academic exchange that has benefited us greatly. Thank you again to the reviewers for their valuable feedback on this manuscript.

Thanks and Regrads,

*Submission 36 authors*

---

### Meta-Review · Area_Chair_7nee · 2024-12-22

**Metareview:**

Summary:
The paper introduces InnerSightNet, an algorithm for analyzing and interpreting the inner workings of deep neural networks through a community-based perspective. The key findings include:
- A three-phase approach  for transforming and analyzing learnable units within neural networks
- Identification of distinct neuron communities with specific functional roles
- Demonstration of how community analysis can improve model interpretability and enable efficient network pruning
- Evidence that certain neuron communities correspond to interpretable features in the input data
- Validation that removing noisy neuron communities can improve model performance

Strengths:
1. Novel Perspective: The community-based approach to analyzing neural networks provides a fresh and insightful way to understand network behavior beyond individual neurons or layers.
2. Well-Structured Framework: The three-phase algorithm combines theoretical foundations with practical implementation in a clear and systematic way.
3. Comprehensive Evaluation: The paper demonstrates the utility of the approach through multiple experiments, including accuracy drop tests and sensitivity analyses.
4. Practical Applications: The method shows promise for network pruning and noise reduction while maintaining performance.
5. Clear Presentation: The methodology is well-explained, particularly through Algorithm 1 and clear visualization of results.

Weaknesses:
1. Scalability Concerns: While the method works well on smaller networks, its computational feasibility for very large networks needs more exploration.
2. Limited Scope: Initial experiments focused mainly on image classification tasks, though authors later demonstrated applicability to other domains.
3. Baseline Comparisons: The paper could have provided more detailed explanations of baseline methodologies for clearer comparison.
4. Runtime Performance: The method shows longer runtime compared to baseline approaches, though this is justified by improved performance.

Reasons for Acceptance:
1. Innovation: The paper presents a novel and well-thought-out approach to neural network interpretation through community analysis.
2. Theoretical Foundation: The mathematical framework is solid and well-justified.
3. Practical Utility: The method demonstrates concrete benefits in terms of network pruning and performance improvement.
4. Comprehensive Response: Authors thoroughly addressed reviewer concerns during the rebuttal period with additional experiments and analyses.
5. Strong Evaluation: The experimental results validate the method's effectiveness across multiple scenarios.

**Additional Comments On Reviewer Discussion:**

During the rebuttal period, reviewers raised several key concerns that were systematically addressed by the authors. Reviewer ymgS sought clarification on mathematical formulations, accuracy drop measurements, and cross-layer community analysis. The authors responded with detailed mathematical explanations and additional cross-layer experiments that satisfied these concerns, leading to an improved score.
Reviewer Rnf1 questioned the clarity of baseline comparisons and scalability to deeper networks. The authors addressed these by providing comprehensive baseline methodology comparisons and demonstrating successful application to ProgressGAN. While Reviewer Rnf1 did not respond to the rebuttal, their initial concerns were thoroughly addressed in the authors' response.
Reviewer 7WnU expressed concerns about scalability to large networks and applicability beyond image classification. The authors responded by showcasing applications to both ProgressGAN and IMDB sentiment analysis, demonstrating the method's broader utility. They also provided detailed justification for the improvements in noise reduction and pruning capabilities.
The authors' thorough responses and additional experiments led to improved scores from two reviewers, reflecting the successful addressing of initial concerns. The comprehensive nature of these responses, combined with the demonstrated broader applicability of InnerSightNet, strongly supports the acceptance decision.

---

### Decision · Program_Chairs · 2025-01-22

Accept (Poster)